# Arrayed CRISPRi and quantitative imaging describe the morphotypic landscape of essential mycobacterial genes

**Timothy J de Wet[1,2]\***, **Kristy R Winkler[1,2]**, **Musa Mhlanga[2,3]**, **Valerie Mizrahi[1,2,4]**, **Digby F Warner[1,2,4]\***

[1]SAMRC/NHLS/UCT Molecular Mycobacteriology Research Unit, Department of Pathology, University of Cape Town,  Cape Town, South Africa; [2]Institute of Infectious Disease and Molecular Medicine, University of Cape Town,  Cape Town, South Africa; [3]Department of Integrative Biomedical Sciences, University of Cape Town,  Cape Town, South Africa; [4]Wellcome Centre for Infectious Diseases Research in Africa, University of Cape Town,  Cape Town, South Africa

**Abstract** *Mycobacterium tuberculosis* possesses a large number of genes of unknown or predicted function, undermining fundamental understanding of pathogenicity and drug susceptibility. To address this challenge, we developed a high-throughput functional genomics approach combining inducible CRISPR-interference and image-based analyses of morphological features and sub-cellular chromosomal localizations in the related non-pathogen, *M. smegmatis*. Applying automated imaging and analysis to 263 essential gene knockdown mutants in an arrayed library, we derive robust, quantitative descriptions of bacillary morphologies consequent on gene silencing. Leveraging statistical-learning, we demonstrate that functionally related genes cluster by morphotypic similarity and that this information can be used to inform investigations of gene function. Exploiting this observation, we infer the existence of a mycobacterial restriction-modification system, and identify filamentation as a defining mycobacterial response to histidine starvation. Our results support the application of large-scale image-based analyses for mycobacterial functional genomics, simultaneously establishing the utility of this approach for drug mechanism-of-action studies.

**\*For correspondence:**
timdewet@gmail.com (TJW);
digby.warner@uct.ac.za (DFW)

**Competing interests:** The authors declare that no competing interests exist.

## Introduction

The acquisition of genomic data continues to exceed significantly the pace at which corresponding functional information can be obtained (*Kemble et al., 2019*). This aphorism applies to *Mycobacterium tuberculosis*, etiological agent of tuberculosis (TB) and the leading cause of death globally from an infectious disease (*WHO, 2019*). Despite considerable activity (*Satta et al., 2018*) in the two decades subsequent to the pioneering release of the first complete mycobacterial whole-genome sequence (*Cole et al., 1998*), the *M. tuberculosis* genome still contains a large number of genes of unknown or hypothetical function (*Mazandu and Mulder, 2012*; *Satta et al., 2018*). Moreover, functional validation is lacking even for many annotated genes, undermining fundamental understanding of mycobacterial metabolic and cellular functions which impact pathogenicity and drug susceptibility. There is consequently a pressing need for tractable, high-throughput approaches that can inform mycobacterial gene function rapidly and at scale.

The most commonly applied functional genomics methods in bacteria – transposon-sequencing (Tn-Seq) (*van Opijnen and Camilli, 2013*) and, increasingly, CRISPR-interference (CRISPRi)-Seq (*de Wet et al., 2018*; *Wang et al., 2018*; *Lee et al., 2019*) – combine pooled mutagenesis with next-generation sequencing, returning quantitative estimates of fitness (via relative abundance) of

**eLife digest** Caused by the microorganism *Mycobacterium tuberculosis,* tuberculosis kills more people around the world than any other infectious disease. *M. tuberculosis* is also becoming increasingly resistant to treatments, which are particularly difficult for patients to complete. The *M. tuberculosis* genome carries about four thousand genes, with several hundred being vital for survival. Finding new ways to fight tuberculosis relies on understanding the exact role of these essential genes, but they are difficult to study in living bacteria.

To investigate this question, de Wet et al. used the related, fast-dividing bacterial species called *M. smegmatis* as a model. Microscopic imaging was combined with CRISPR-interference – a method that temporarily disrupts expression of a specific gene – to examine how blocking an essential gene would affect the shape of the living microorganism.

Experiments were conducted on a collection of 270 mutants, capturing single-cell data for hundreds of thousands of live bacteria. To analyze the data, a computational pipeline was built, which automatically clustered similar-shaped bacteria. These groups, or 'phenoprints', brought together genes of known and unknown roles; this indicated that these genes participate in similar biological networks – and, if unknown, hinted at their function.

Finally, targeting essential genes with CRISPR-interference often yielded the same shape changes as blocking their encoded proteins with antibiotics. This suggests that phenoprints could be useful to understand the mode of action of potential new tuberculosis treatments. When applied to *M. tuberculosis* and other deadly bacteria, the approach developed by de Wet et al. might speed up drug development.

mutants in a particular growth condition. Common to these approaches is that they report on the capacity of the cells to produce biomass, thereby excluding other – potentially subtler – measures of the physiological consequences of target gene disruption. To overcome this limitation, some studies have utilized novel combinations of pooled approaches with cellular stains and sorting (*Rego et al., 2017*; *Baranowski et al., 2018*), or with imaging and in situ sequencing (*Camsund et al., 2020*) to provide alternative readouts of genetic function. However, these methods are restricted in their resolution and throughput.

As for many bacterial pathogens, mycobacterial functional genomics has focused primarily on the identification of essential genes, employing Tn-based investigations under a variety of conditions (*Sassetti et al., 2003*; *DeJesus et al., 2017*), in different strain (*Carey et al., 2018*) and mutant (*Kieser et al., 2015*) backgrounds, and in models of infection (*Sassetti and Rubin, 2003*; *Rengarajan et al., 2005*). The development of genetic tools for conditional knockdown mutagenesis via targeted gene silencing (utilizing engineered hypomorphic strains carrying promoter replacement and/or targeted protein-degradation mutations) has subsequently enabled gene-by-gene explorations of essential gene function (*Kim et al., 2011*; *Kim et al., 2013*; *Schnappinger and Ehrt, 2014*). However, while these approaches have ensured increasingly refined (conditional) essentiality predictions (*DeJesus et al., 2017*), there is still an absence of whole-cell functional data for the majority of essential genes.

Here, we couple key recent advances in mycobacterial CRISPRi (*Rock et al., 2017*) with quantitative microscopic imaging (*Ducret et al., 2016*) in developing a platform enabling whole-cell characterization of essential gene knockdown in the non-pathogenic model mycobacterium, *M. smegmatis*. Leveraging available gene essentiality and CRISPRi guide efficacy data (*de Wet et al., 2018*), we describe the construction of an arrayed collection of 276 validated CRISPRi mutants targeting essential *M. smegmatis* homologs of *M. tuberculosis* genes. Applying high-throughput quantitative imaging to the arrayed library, we implement a bespoke analytic pipeline to probe essential gene function at scale. From the 'phenoprints' generated via this approach, we demonstrate the potential for preliminary assignment of gene function and illustrate the capacity to analyze metabolic and macromolecular synthesis pathways using clustering analyses. Notably, these observations provide evidence supporting a mycobacterial restriction-modification system, and identify filamentation as a defining mycobacterial response to histidine starvation but not to other amino acid auxotrophies. Finally, we demonstrate the potential utility of this approach in elucidating antimicrobial mechanism-

of-action (MOA), supporting its potential incorporation as a complementary tool in current TB drug discovery pipelines. Consistent with our intention to generate a community-accessible resource, all data from the screen are made available via an interactive database (https://timdewet.shinyapps.io/MorphotypicLanscape/).

## Results

### Construction of an arrayed CRISPRi library targeting essential mycobacterial genes

We aimed to construct an arrayed library of inducible CRISPRi mutants (*Figure 1A*) targeting essential *M. smegmatis* genes (*de Wet et al., 2018*). To this end, we leveraged an optimized mycobacterial CRISPRi system (*Rock et al., 2017*) and previously generated genome-wide CRISPRi-Seq data (*de Wet et al., 2018*) from which we were able to identify the set of highest efficiency single-guide (sg)RNAs targeting 294 essential *M. smegmatis* genes with direct *M. tuberculosis* homologs (*Supplementary file 1*).

CRISPRi produces polar effects on downstream genes in polycistronic operons (*Rock et al., 2017*). To ascertain operonic structure directly, we utilized published data on the transcriptional landscape of wild-type *M. smegmatis* mc$^2$155 during exponential growth in standard laboratory media (*Martini et al., 2019*). Of our targeted set of 294 genes, 229 genes (~78%) were not located in identified operons (*Supplementary file 2*). For these genes, it was expected that any knockdown phenotype should solely reflect the impact of silencing the targeted essential gene. Of the remaining 65 genes within operons, 20 possessed downstream transcriptional start sites which would be expected to abrogate polar effects. This left 34 genes whose genomic context was likely to complicate the interpretation of any knockdown phenotype. We nevertheless considered this group of 34 genes worth pursuing given our ability to evaluate the results in the context of the corresponding essentiality calls from Tn-Seq (*Dragset et al., 2019*) and in comparison to phenotypes observed for related (or 'shared pathway') genes.

Following large-scale cloning, all 294 CRISPRi constructs were electroporated into an *M. smegmatis* ParB-mCherry reporter mutant (*Santi and McKinney, 2015*). This strain was chosen as background to enable visualization of the *oriC* proximate region of the chromosome during imaging assays, thereby providing information about chromosome location dynamics and copy number as a function of essential gene knockdown. To validate the identity and essentiality of each *M. smegmatis* mutant strain, we performed Sanger sequencing of the inserted sgRNA and whole-cell spotting assays of ATc sensitivity. Cloning was repeated for mutants which failed the validation screen. This process yielded 276 validated CRISPRi mutants (*Figure 1B*, *Figure 1—figure supplement 1*), 93% of the initial target set. According to a review of the literature (*Supplementary file 3*), approximately 90% of the strains in our library lacked whole-cell morphological characterization in either *M. tuberculosis* or *M. smegmatis*. Furthermore, almost 40% of the targeted genes and their protein products had no biochemical or structural information (*Supplementary file 3*).

### A quantitative imaging pipeline for mycobacteria

High-throughput, quantitative imaging has been productively utilized in a number of bacterial systems to rapidly characterize the impact of genetic alterations on cellular function (*Peters et al., 2016*; *Liu et al., 2017*; *Campos et al., 2018*) or to determine antimicrobial MOA (*Nonejuie et al., 2013*). Until very recently (*Smith et al., 2020*), an equivalent approach had not been described for mycobacteria. Therefore, on initiating this study, we chose to investigate the use of imaging to phenotype the CRISPRi mutant library following ATc-dependent transcriptional silencing. This required the development of tools for the extraction of large-scale quantitative data describing mycobacterial cell morphology.

To determine the optimal duration of ATc exposure, we performed time-lapse microscopy of knockdown mutants of three well-characterized essential genes (*Videos 1* and *2*) – the cell-division mediator, *ftsZ* (*Dziadek et al., 2003*), the elongasome anchor, *wag31* (*Kang et al., 2008*), and the inosine monophosphate dehydrogenase, *guaB2* (*Singh et al., 2017*; *Park et al., 2017*). Time-lapse microscopy established that fully penetrant phenotypes were manifest at 18 hr post ATc exposure for all three genes. This represents approximately 6–7 doubling times of the wild-type *M. smegmatis*

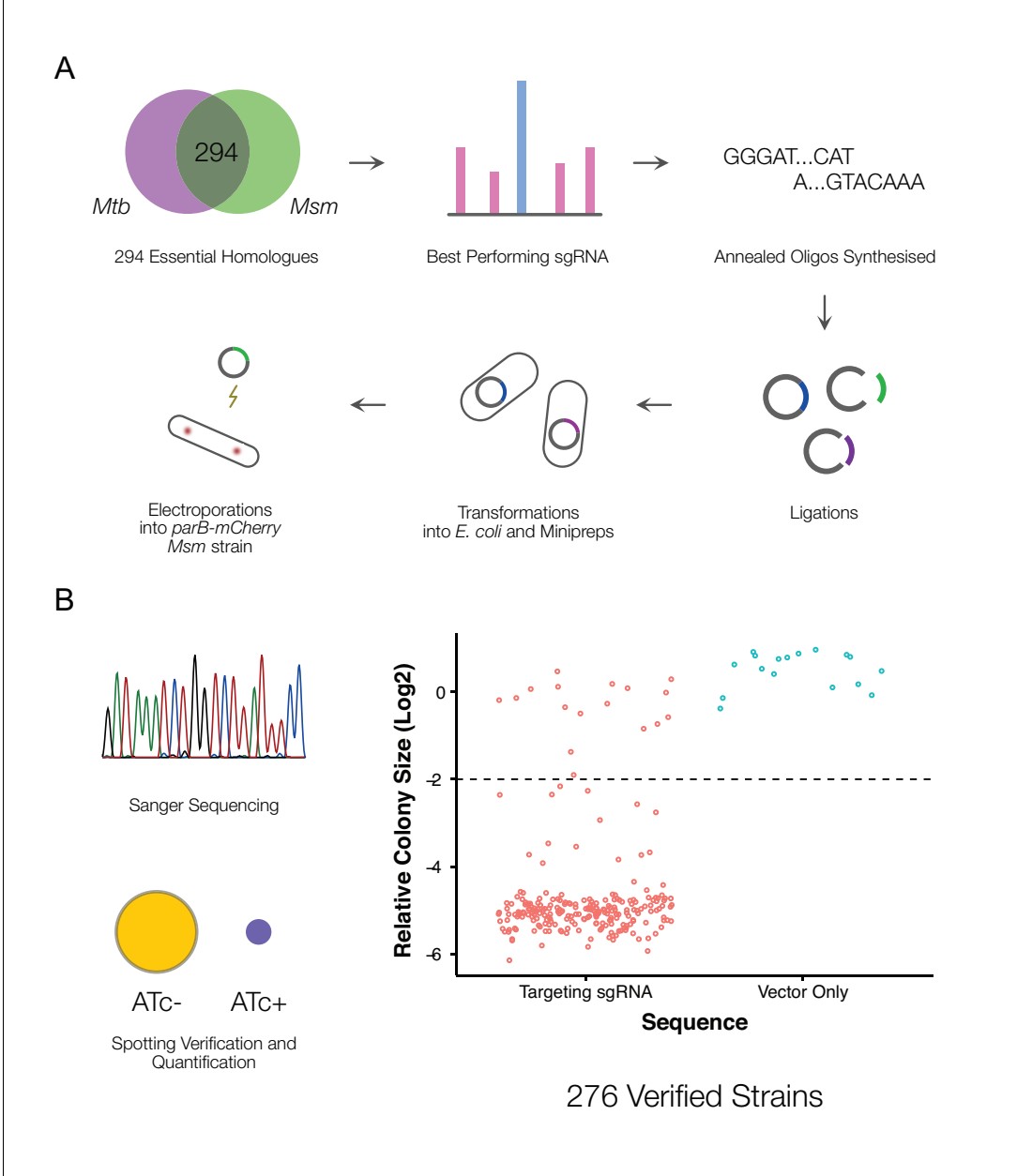

**Figure 1.** A CRISPRi library targeting essential *M. smegmatis* homologs of *M. tuberculosis* genes. (**A**) An arrayed CRISPRi library was designed to target 294 essential *M. smegmatis* genes. For each gene, the highest efficiency sgRNA was identified from a previous pooled CRISPRi-Seq screen (*de Wet et al., 2018*) and synthesized as an annealed oligonucleotide. Cloning was performed at scale, followed by electroporation into an *M. smegmatis* ParB-mCherry reporter strain (*Santi and McKinney, 2015*). (**B**) A total of 276 sequence-verified transformants produced a greater than twofold decrease in colony size (*Kritikos et al., 2017*) when spotted on 7H10 agar containing the inducer, anhydrotetracycline (ATc) compared to the same cells spotted onto solid medium without ATc, confirming ATc-dependent growth inhibition.

The online version of this article includes the following figure supplement(s) for figure 1:

**Figure supplement 1.** Numbers of strains used at each point of this study.

mc²155 parental strain (*Logsdon et al., 2017*), or a dilution factor of pre-existing protein of approximately 64 to 128 ($2^6$ - $2^7$)-fold. For both *wag31* and *ftsZ* mutants, lytic cell death was observable after the 18 hr time-point. While *guaB2* knockdown did not produce as distinctive a visual phenotype as the other two genes, the formation of minicells and a decline in growth rate was evident from 18

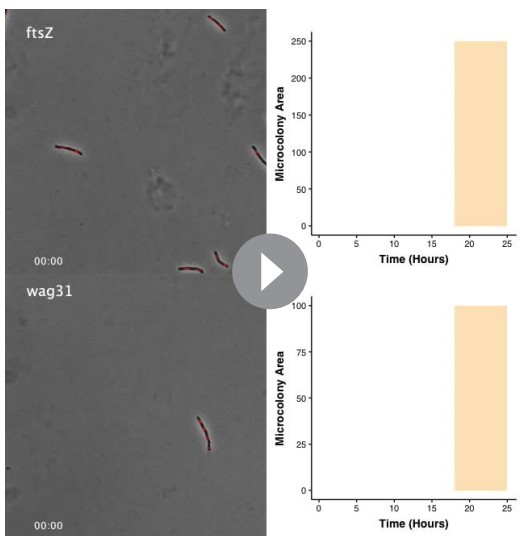

**Video 1.** Time-lapse microscopy of *ftsZ* and *wag31* silencing by CRISPRi.

https://elifesciences.org/articles/60083#video1

hr. Although this test set comprised only three mutants, we were encouraged that, for these essential genes with distinct cellular functions, the 18 hr timepoint appeared suitable to ensure sufficient knockdown and dilution of pre-existing protein to yield a detectable phenotype while maintaining suitable cell numbers for imaging. We therefore proceeded to apply the same experimental conditions in phenotyping the full library, recognizing that the pragmatic utilization of a single, 'terminal' endpoint might not be optimal for every mutant in the collection.

Adopting the 18 hr CRISPRi duration, we derived a protocol for medium-throughput imaging of batches of 24 strains (*Figure 2A*). Exponential-phase cultures were treated with ATc overnight for 18 hr prior to spotting onto 2% agarose pads for semi-automated imaging. Phase contrast and fluorescence images of ParB-mCherry localizations were captured for 263 of the 276 mutant strains. Additionally, an empty vector control strain was treated and repeatedly imaged in the same manner. To confirm the reproducibility of data obtained from the single-timepoint, single-replicate imaging workflow, we selected 137 strains for re-imaging on separate days.

Image analysis tools such as MicrobeJ (*Ducret et al., 2016*) or Oufti (*Paintdakhi et al., 2016*) extract detailed quantitative descriptions of bacterial morphology and protein localization (*Figure 2—figure supplement 1*). However, automated image analysis is complicated by out-of-focus objects, or cells that are too close together to separate adequately – a particular concern for mycobacteria with their proclivity to clump (*Cheng et al., 2014*). To overcome these complications, machine learning-based classifiers have been utilized in *E. coli* for post-processing clean-up (*Campos et al., 2018*).

To extract quantitative descriptions of mutant morphologies and ParB-mCherry localizations across our imaging dataset, we utilized the ImageJ package, MicrobeJ (*Ducret et al., 2016*), with a machine learning-based post-processing clean-up (*Figure 2B*). For this purpose, we trained an Averaged Neural Network with 22,936 manually classified objects, sampled from across our imaging dataset. Receiver Operator Characteristic (ROC) analysis of our classifier produced an Area Under the Curve (AUC) of 0.952 when applied to a reserved test set (*Figure 2C*). Our curated dataset contains morphological descriptions for 163559 cells across all 263 imaged mutant and empty vector control strains, with a mean of 568 cells per mutant (ranging from a minimum of 24 to a maximum of 3861 cells). Reassuringly, when comparing the mean cell lengths of the 137 replica-imaged knockdown mutants following ATc induction, the imaging and analytic pipeline showed high reproducibility (*Figure 2D*, r = 0.88, Pearson's).

## An atlas of morphological changes consequent on essential gene silencing

There are relatively few published descriptions of the morphological impacts of essential gene silencing (*Supplementary file 3*). We aimed to use our extracted imaging data to generate a comprehensive repository of morphological changes following CRISPRi-mediated essential

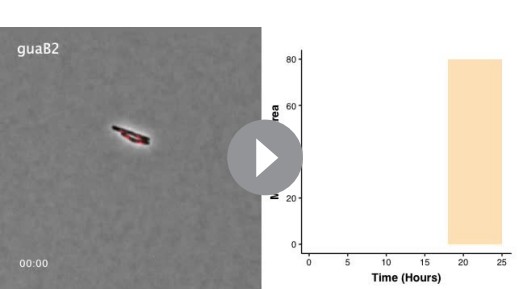

**Video 2.** Time-lapse microscopy of *guaB2* silencing by CRISPRi.

https://elifesciences.org/articles/60083#video2

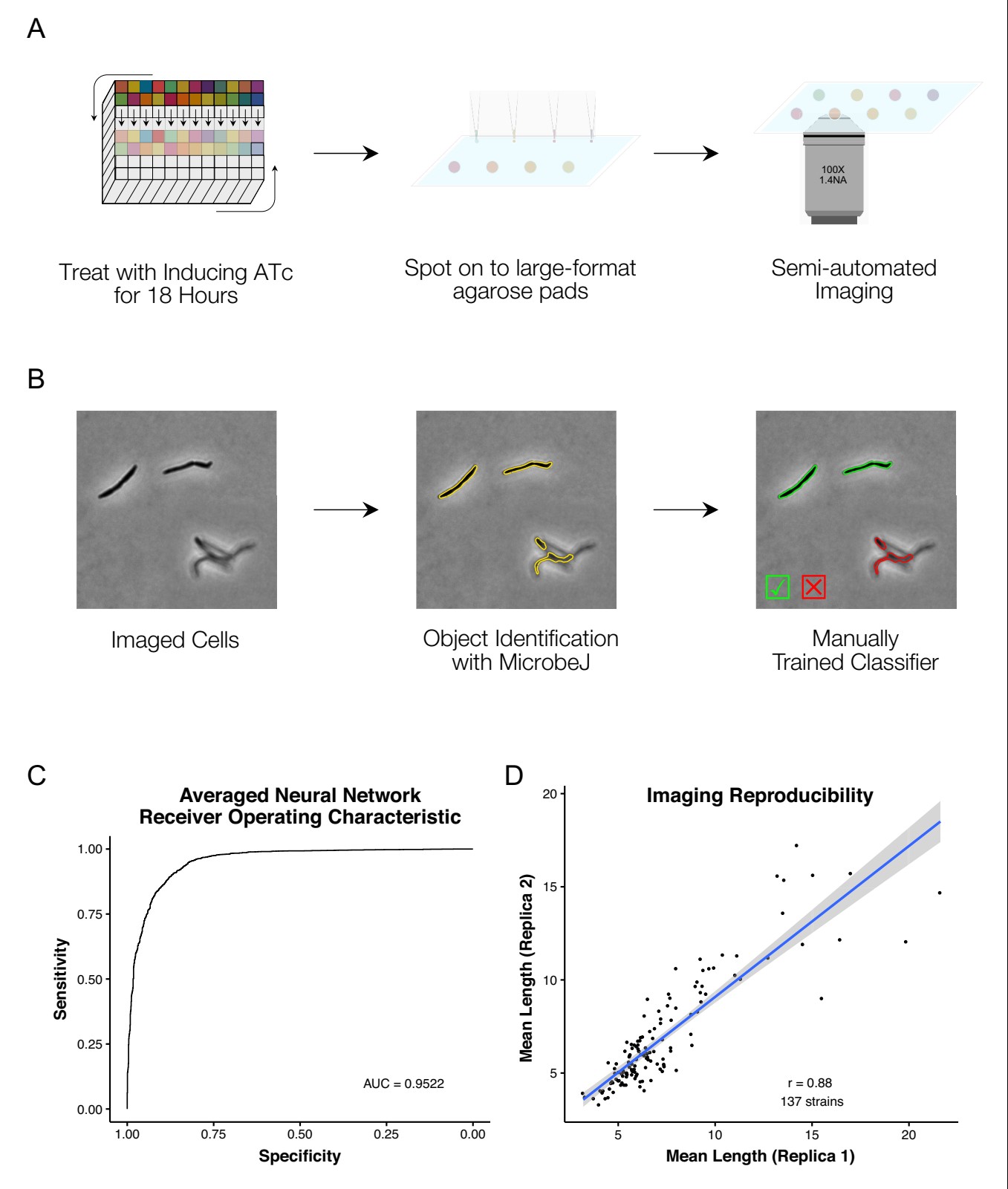

**Figure 2.** A high-throughput, quantitative CRISPRi-imaging pipeline for mycobacteria. (A) Cells were exposed to ATc inducer for 18 hr before spotting onto large-format agarose pads for semi-automated imaging. (B) Image processing in MicrobeJ (*Ducret et al., 2016*) was combined with a manually trained Averaged Neural Network classifier to extract quantitative descriptions of bacterial morphologies and ParB protein localizations for 163559 cells across 263 gene-specific CRISPRi mutants and 27 empty vector controls. (C) Classifier performance was measured by Receiver Operating Characteristic

*Figure 2 continued on next page*

*Figure 2 continued*

(ROC) Area Under the Curve (AUC), returning good performance metrics. (**D**) Mean cell lengths were compared for 137 strains imaged as biological replicates on two separate occasions, and showed high reproducibility (r = 0.88, Pearson's).

The online version of this article includes the following figure supplement(s) for figure 2:

**Figure supplement 1.** Features extracted by MicrobeJ (*Ducret et al., 2016*) and utilized in downstream analysis.

**Figure supplement 2.** Consistency of phenotypes with varying sgRNAs.

**Figure supplement 3.** Comparisons of cell classifier models.

gene knockdown in *M. smegmatis*. For each gene, a nearest-neighbor approach was applied to identify the representative morphotype which was in turn incorporated into an atlas of morphological changes consequent on essential gene silencing (*Figure 3*).

Visual inspection of the atlas revealed distinctive alterations in morphology in a large number of knockdown mutants. Moreover, it appeared that silencing certain cellular functions or pathways produced consistent morphological changes. In general, the responses could be broadly categorized into either elongation – occasionally with bulging and/or branching – or shortening, with a variety of alterations in cell width and roundness. For example, components of DNA metabolism (*dna-* prefix genes), the divisome (*fts-* genes), and protein translation (*rpl-* and *rps-* genes) produced cellular elongation. To our surprise, a filamentation response was also observed for components of the histidine biosynthesis pathway (*his-* genes), but not for other amino acid biosynthesis pathways. In contrast, components of cell-wall homeostasis – including peptidoglycan (*ponA1*), arabinogalactan (*aftA*) and mycolic acid synthesis (*inhA* and *mmpL3*) – were associated with distinctive shortening of cells, often in combination with bulging.

To quantitate these observations, we adapted an approach developed for *E. coli* (*Campos et al., 2018*). Single-cell data were reduced to normalized, gene-level descriptors of morphology. For each mutant, the mean and coefficient of variation (CV) of each morphological feature were extracted and normalized to the distribution of empty vector control samples with a Z-score transformation (*Figure 4—figure supplement 1A*). For a particular mutant and feature, the Z-score reflects how many standard deviations the mutant feature is away from the mean of the control strains. That is, it provides a measure of the extent to which transcriptional inhibition of the targeted essential gene impacts a specific morphological characteristic. In total, 206 (78%) strains returned at least one Z-score >3, or <-3. Certain gross changes in morphology were more frequently observed than others (*Figure 4—figure supplement 1B*). For example, many mutants exhibited marked increases in mean curvature, roundness and feret (the calliper diameter). Conversely, few mutants exhibited major alterations in the number of ParB foci (maxima) per cell, suggesting that the number of replication origins (*oriC*s) per cell – and, by implication, ploidy – was maintained even under essential gene knockdown.

To test for associations between morphological changes and functional classes of genes, we extracted clusters of orthologous groups (COG) annotations for all 276 essential genes before testing for statistical enrichment with changes in morphology. For each morphological feature, we tested for COG enrichment of the mutants with Z-scores > 3 or<-3. This analysis revealed that statistical enrichments were largely consistent with visual inspection of the cell atlas (*Figure 4—figure supplement 2*). For example, the filamentation phenotype of *fts-*, *rpl-* and *dna-* genes was reflected in the enrichment of cell cycle control (D), Translation (J) and DNA replication and repair (L) COGs with increases in length (means and CVs).

## The morphotypic landscape of essential gene silencing in mycobacteria

In a complementary approach, we visualized the mean population Z-scores for key morphological features by COG (*Figure 4*). Again, consistent with the cell atlas and enrichment results, specific COGs (*Supplementary file 3*) produced distinct morphological signatures (or 'phenoprints'). Together, these results highlighted the utility of multi-parameter morphological descriptors for characterizing mutants and, moreover, signaled the potential to link genes according to whole-cell phenotype independent of known or predicted (annotated) function. Based on prior cytological profiling studies in eukaryotic (*Caicedo et al., 2017*) and prokaryotic (*Nonejuie et al., 2013*; *Huang, 2015*; *Campos et al., 2018*) systems, we applied an unsupervised learning approach that combined

**Figure 3.** An atlas of morphological changes consequent on essential gene silencing in *M. smegmatis.* Following imaging and extraction of quantitative data on cellular morphology, an approximate nearest-neighbor approach was utilized to identify a single representative cell which was closest to the mean values of the measured morphological features for each specific (clonal) mutant population. Representative cell contours and ParB localization patterns were extracted from MicrobeJ and utilized to assemble the atlas of morphological changes. Cells are grouped and colored according to downstream clustering (*Figure 5*).

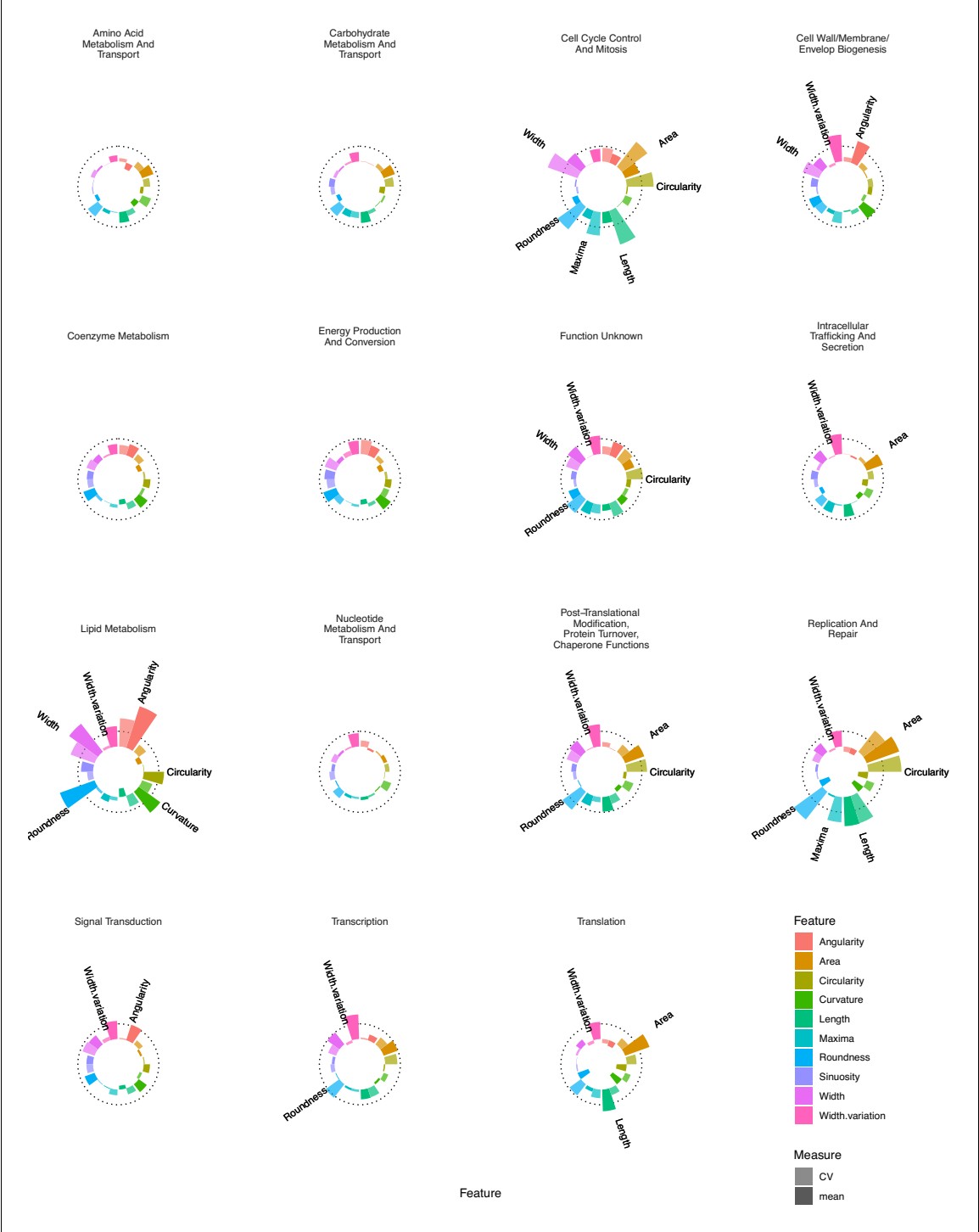

**Figure 4.** Essential gene silencing produces distinctive phenoprints of genetic function. Mean Z-scores were calculated and visualized for all genes assigned to a particular COG, highlighting distinctive changes in morphology. The dotted line indicates a Z-score of 3. Selected features with mean Z-scores > 3 are labeled.

The online version of this article includes the following figure supplement(s) for figure 4:

**Figure supplement 1.** The morphological impact of essential gene silencing.

**Figure supplement 2.** COG enrichment identifies associations between genetic function and morphological changes.

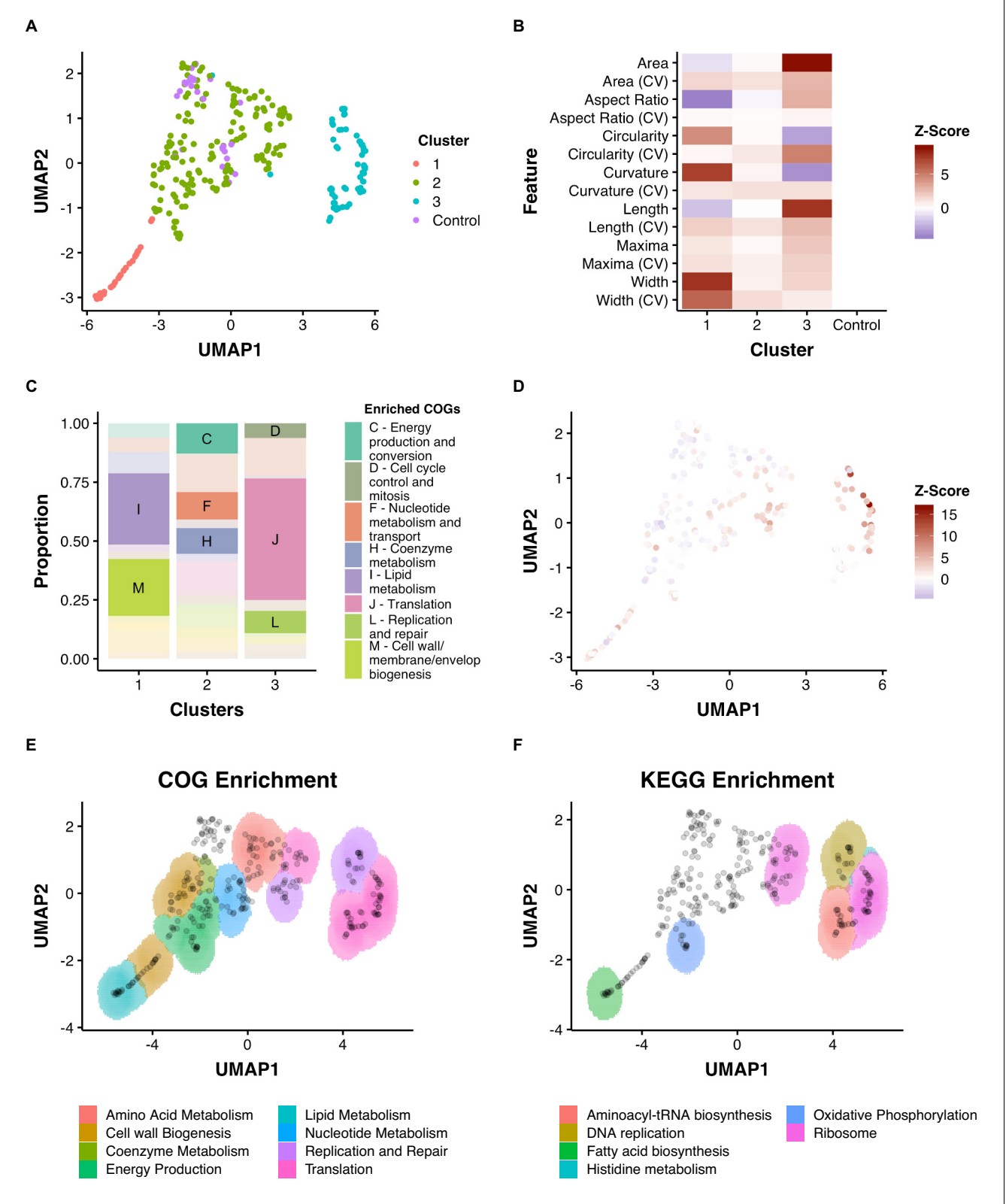

**Figure 5.** Dimensionality reduction reveals clusters of morphological change that associate with function. (A) A combination of UMAP dimensionality reduction (**Lel et al., 2018**) and hdbscan clustering (**Lel et al., 2017**) was applied to processed morphological data. Parameters were optimized to ensure a uniform clustering of control strains (MinPts = 12). A consistent seed was used for visualization. (B) The mean Z-Scores of each cluster highlights the dominant features determining clustering. (C) Clusters are enriched for certain COGs. (D) Overlaid Z-score data for mean aspect ratio,
*Figure 5 continued on next page*

*Figure 5 continued*

length, maxima, and width demonstrate heterogeneity within clusters. (E) and (F) An adapted SAFE (*Baryshnikova, 2016*) approach demonstrates that functionally enriched sub-clusters are present in UMAP space.

The online version of this article includes the following figure supplement(s) for figure 5:

**Figure supplement 1.** Distances in UMAP space can reflect relationships of biological relevance.

dimensionality reduction with clustering of our multidimensional dataset. For this, we utilized uniform manifold approximation and projection (UMAP) (*Lel et al., 2018*) and optimized hyper-parameters to produce consistent visual clusters of our control strains in combination with density-based clustering using hdbscan (*Lel et al., 2017*; *Figure 5*).

This process produced a two-dimensional UMAP space consisting of three reproducible clusters (*Figure 5A*). The scale of different morphological features varied across the 2-dimensional space with visible patterns, such as regions of decreased length, or increased width (*Figure 5D*). To explore the three clusters, we tested for COG enrichment and visualized mean feature Z-scores for each cluster (*Figure 5B and C*). Cluster 1, which was enriched for lipid metabolism (I) and cell-wall metabolism (M), was associated with increases in width and curvature and decreases in area and aspect ratio. Conversely, Cluster 3, which was associated with translation (J) and DNA replication and repair (L), was characterized by increases in area and length. Cluster 2, which contained the control, non-targeting strains, was enriched for energy production and conversion (C) and nucleotide metabolism (F), highlighting the comparatively small impact that silencing of these genes has on cellular morphology.

To analyze further the spatial arrangements within the three large clusters, and across UMAP space, we adapted a spatial analysis of functional enrichment (SAFE) approach (*Baryshnikova, 2016*; *Campos et al., 2018*) to identify sub-clusters of either COG or KEGG enrichment. This yielded distinct sub-clusters that were not originally apparent (*Figure 5E and F*). For example, Cluster 1 contained regions enriched for either lipid metabolism (I) or the cell wall (M), while cluster three could be subdivided into a region enriched for DNA metabolism (L), ribosomal genes (K) and genes involved in aminoacyl-tRNA biosynthesis (by KEGG enrichment). Using this approach, even cluster two could be subdivided into functionally enriched regions despite the apparent similarity of these strains (on visual inspection) to wild-type morphologies.

As UMAP maintains the essential topological structure of data, we wondered if biological connections would be reflected in the distances between genes in UMAP space. Given the acknowledged propensity of CRISPRi to produce polar effects (*Rock et al., 2017*), we predicted that these would be reflected in the Euclidean distance between operonic genes. Consistent with our hypothesis, genes in verified operons (*Martini et al., 2019*) were located more proximally than random pairs of genes (*Figure 5—figure supplement 1*). Similarly, predicted protein–protein interaction pairs (*Cong et al., 2019*) were found to be non-randomly proximate (*Figure 5—figure supplement 1*).

To demonstrate further the utility of UMAP space for understanding the link between gene function and morphology, we performed more granular analysis of manually annotated functional groupings. Again, consistent clustering of many genes with similar biological functions was observed (*Figure 6*), a striking finding given that the close associations in UMAP space were driven by similarities in cytological characteristics (phenoprints) independent of annotated gene function.

The observation that subtle morphological changes appeared to associate with particular genetic functions implied the potential to utilize the pipeline and mutant library to elucidate biological relationships and as an additional tool in MOA studies. Four examples are presented below which illustrate the application of this approach to explore (i) putative genetic function, (ii) pathway-specific phenotypes, (iii) macromolecular biosynthetic phenotypes, and (iv) chemical-genetic approaches to antimycobacterial MOA identification.

## Exploring gene function: a possible mycobacterial restriction-modification system

Morphological profiling has the capacity to identify mutants with unexpected phenotypes, providing a preliminary phenotypic characterization which can guide focused downstream investigations toward assigning gene function. An example illustrating this possibility is *MSMEG_3213*, which is

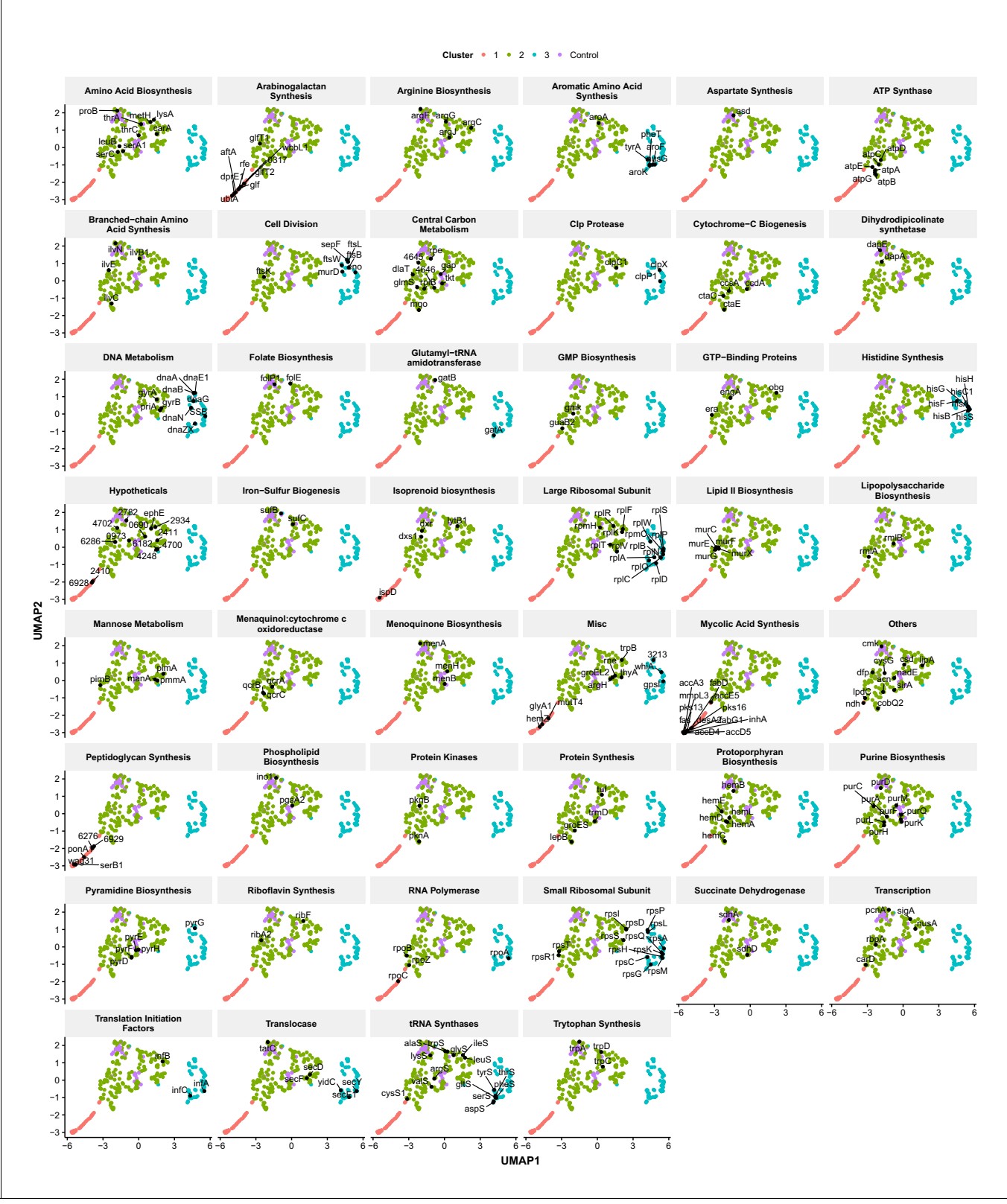

**Figure 6.** Functionally related genes are frequently found associated in UMAP space. Genes were manually classified following a review of the literature (*Supplementary file 3*) and visualized in UMAP space. Numbers represent *M. smegmatis* accession numbers.

annotated as a putative DNA methylase but lacks biological validation. The *M. tuberculosis* homolog, *Rv3263*, has been implicated in bacillary fitness during hypoxia (*Shell et al., 2013*) but, unlike its *M. smegmatis* homolog, is not included in the essential gene set in the pathogen (*DeJesus et al., 2017*). In our imaging pipeline, *MSMEG_3213* produced a marked filamentation phenotype, clustering with components of DNA replication and repair in UMAP space (*Figure 7A and B*). Utilizing the ParB-mCherry marker, we noted a significant disturbance in normal cell-cycle progression, consistent with the interpretation that knockdown of *MSMEG_3213* disrupted DNA replication (*Figure 7C*). Moreover, Nanopore-based RNA-Seq confirmed the specificity of *MSMEG_3213* knockdown, eliminating potential off-target effects on downstream or upstream genes (*Figure 7D*). Notably, the whole-genome transcriptional profile on *MSMEG_3213* knockdown showed marked overlap with the well-characterized DNA damage (SOS) regulon (*Davis et al., 2002*; *Boshoff et al., 2003*) induced by treatment with genotoxins such as the DNA crosslinking agent, mitomycin C (MMC, *Müller et al., 2018*; *Figure 7E*).

Based on this intriguing observation, we hypothesized that, since depletion of the methylase appeared to trigger a DNA damage response, *MSMEG_3213* might be part of a restriction-modification (R-M) system (*Loenen et al., 2014*). Following an extensive bioinformatic search, this conclusion gained further credence by the observation that REBASE (*Roberts et al., 2015*) lists *MSMEG_3213* as putative DNA methylase of a type II R-M system, with *MSMEG_3214* the associated restriction endonuclease (*Figure 7F*). We reasoned that, if *MSMEG_3213-MSMEG_3214* does constitute a cryptic R-M system, knockdown of *MSMEG_3214* should render *MSMEG_3213* non-essential. To test this, a dual-targeting CRISPRi construct was generated to enable simultaneous silencing of both genes. Knockdown of *MSMEG_3214* alone produced no growth phenotype whereas dual silencing of *MSMEG_3213* and *MSMEG_3214* alleviated the lethal impact of *MSMEG_3213* depletion (*Figure 7G*). Further biochemical and/or functional characterization is required before MSMEG_3213 can be definitively assigned as methylase; however, the evidence derived here from morphological profiling, transcriptomic profiling and combinatorial CRISPRi strongly support the identification of a predicted Type II R-M system in *M. smegmatis*.

Other examples supporting the utility of morphological profiling to inform single-gene functional analyses arose during the course of this work (*Figure 7—figure supplement 1*). For example, the *C. glutamicum* homolog of *MSMEG_0317* is required for lipomannan maturation and lipoarabinomannan (LM/LAM) synthesis (*Cashmore et al., 2017*). We observed that *MSMEG_0317* clustered closely with genes involved in arabinogalactan synthesis and localized to the cell wall (*Figure 7—figure supplement 1B*). It was pleasing, therefore, when a separate study emerged suggesting that *MSMEG_0317* was involved in the transport of LM/LAM (*Gupta et al., 2019*). In another example, the transcriptional regulator, *whiA*, which is involved in sporulation in *Streptomyces* (*Bush et al., 2013*), appears to play a key role in the mycobacterial cell cycle, clustering with other components of cell division (*Figure 7—figure supplement 1C*). Furthermore, *MSMEG_6276* – a putative mur ligase – clusters closely with the peptidoglycan synthesis protein, *mviN* (*Gee et al., 2012*), and exhibits strong homology to *murT/gatD* from *S. pneumoniae* (*Morlot et al., 2018*; *Figure 7—figure supplement 1D*). In combination, these additional examples support the utility of image-based profiling for informing or validating hypothetical or predicted gene function – particularly those involved in cell-wall metabolism or DNA metabolism and cell-cycle regulation.

## Identifying pathway-specific phenotypes: histidine auxotroph filamentation

A feature of the library is that many metabolic pathways are represented by multiple mutants, allowing for pathway-level analyses of metabolic function. *M. smegmatis* possesses a full complement of genes for the biosynthesis of the essential amino acid, L-histidine (*Figure 8A and B*). In the absence of histidine supplementation, the majority of the *his-* prefix genes are predicted to be essential, but most have not been validated individually (*Lunardi et al., 2013*). Surprisingly, CRISPRi-mediated knockdown of histidine biosynthesis genes produced a filamentation phenotype that was tightly clustered in UMAP space (*Figure 8C*) and, among all the amino acid knockdowns, appeared unique to histidine. The possibility existed that the operonic arrangement of some genes in the pathway (*hisB*, *hisC1*, *hisH*, *hisA*, *hisF*, *hisI*; *hisE*, *hisG*) had confounded interpretation of the CRISPRi phenoprints as a result of polar effects. However, our transcriptional data and those of others (*Martini et al., 2019*) appeared to eliminate this possibility: *hisD*, *hisC*, *hisB*, *hisH* and *hisF* possess nested promoters and

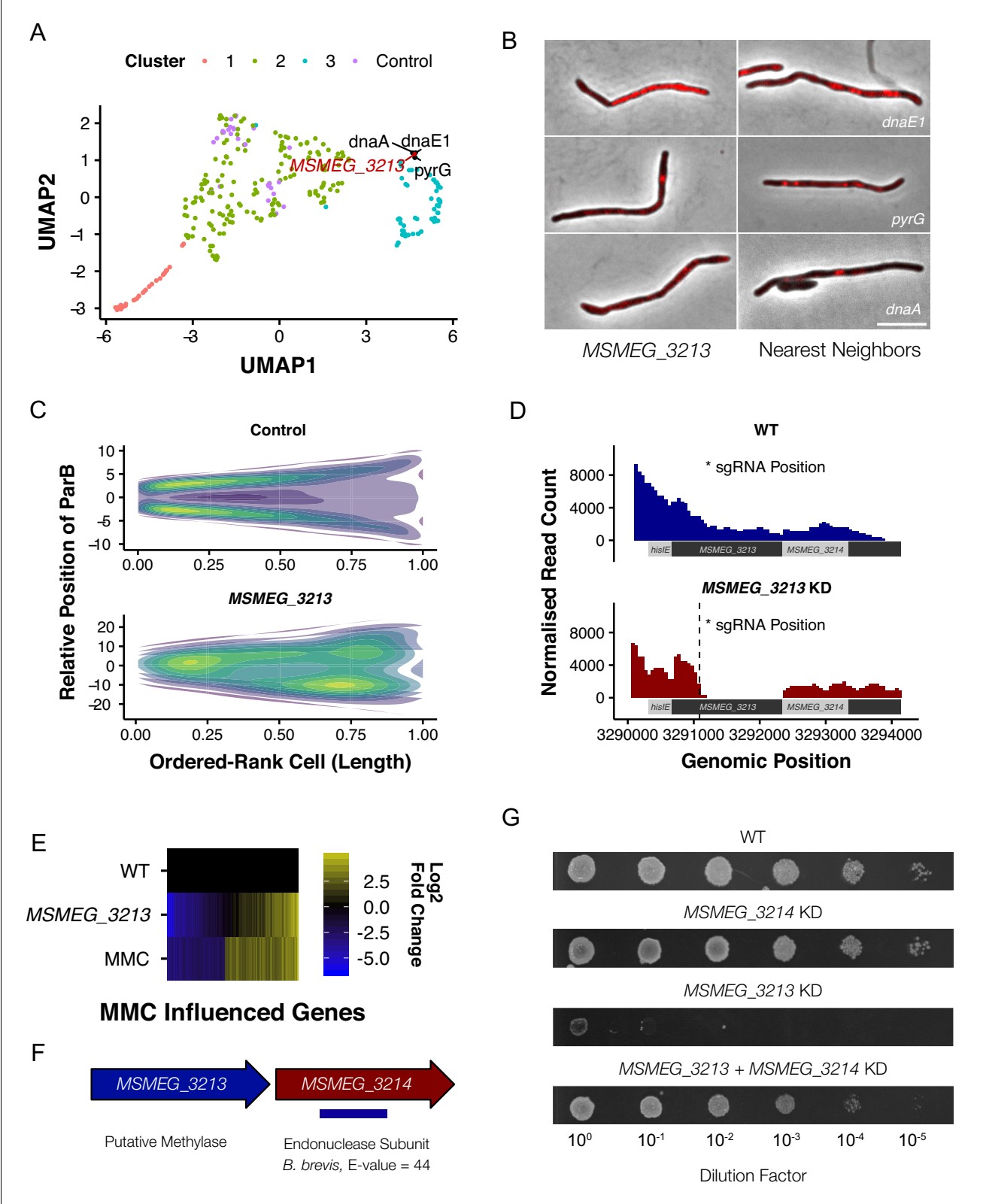

**Figure 7.** *M. smegmatis* encodes a previously undescribed restriction-modification system. (**A**) MSMEG_3213, a putative DNA methylase, is associated with genes involved in DNA replication and repair in UMAP space. (**B**) Knockdown of MSMEG_3213 leads to cellular filamentation. The morphological nearest neighbors are dnaE1, pyrG and dnaA. (**C**) Consensus heatmaps of ParB-mCherry localization demonstrate that oriC positioning is disrupted in the MSMEG_3213 knockdown mutant. (**D**) Nanopore-based RNA-Seq confirmed that knockdown of MSMEG_3213 was specific to the targeted sgRNA

*Figure 7 continued on next page*

*Figure 7 continued*

position. (E) MSMEG_3213 produces a transcriptional response comparable to treatment with the DNA damaging agent, mitomycin C (MMC). (F) MSMEG_3213 is located upstream of MSMEG_3214, a gene with weak homology, according to HHpred (*Hildebrand et al., 2009*), to an endonuclease subunit (REBASE: 3098 BbvCI). (G) Lethality of MSMEG_3213 knockdown is suppressed by simultaneous CRISPRi-mediated knockdown of MSMEG_3214.

The online version of this article includes the following figure supplement(s) for figure 7:

**Figure supplement 1.** Morphological profiling informs gene function.

---

are therefore likely to be impervious to CRISPRi-mediated silencing of upstream genes. Moreover, CRISPRi knockdown of *hisG* and *hisS*, which are distally located in the *M. smegmatis* genome, produced the same phenotype, suggesting that filamentation might be a consistent, and previously unreported, consequence of disruptions to histidine biosynthesis.

To test the specificity of this phenotype, we supplemented ATc-containing media with L-histidine. The reversal of the growth inhibitory phenotype verified all *his* knockdowns as histidine auxotrophs, with the notable exceptions of *hisA* and *hisS*. Previous reports indicate that *hisA* likely encodes a bifunctional HisA/TrpF enzyme (*Due et al., 2011*). Consistent with this annotation, growth of the *hisA* CRISPRi mutant was rescued in medium supplemented with tryptophan (*Figure 8D*). Intriguingly, for all *his* knockdowns other than *hisS* (encoding the histidine tRNA synthetase), histidine supplementation reversed the filamentation phenotype (*Figure 8E*). This was also the case for *hisA*, even when the lack of tryptophan impaired growth. Morphologically, the filaments observed under histidine starvation differ from those which result from depletion of components of the divisome: that is, the *his* mutants display minimal branching. Moreover, staining of peptidoglycan with the D-alanine analogue, NADA (*Botella et al., 2017*), failed to resolve any clear septa (*Figure 8—figure supplement 1*). The mechanism underlying the filamentation response to histidine deficiency remains unclear, but likely reflects translation defects resulting from uncharged tRNAs. Further work is required to establish this definitively, but the observations reported here nevertheless support the potential utility of the CRISPRi-imaging pipeline to discover, and then validate, pathway-specific phenotypes.

## Validating macromolecular pathway phenotypes: morphotypic consequences of disrupting mycolic acid biosynthesis

Mycobacteria possess a multilayered cell envelope, the outer mycolic acid layer of which is synthesized via a well-characterized pathway containing a number of established and experimental drug targets (*Jankute et al., 2015*; *Figure 9A*). CRISPRi-mediated inhibition of the components of this pathway produced consistent changes in cellular morphology (*Figure 9B and C*), with the majority of genes implicated in mycolic acid synthesis clustering tightly together in 2-dimensional UMAP space (*Figure 9B*). Notably, the fact that clustering was observed despite the location of these genes throughout the chromosome appeared to eliminate polar effects as a potential confounder.

By superimposing our imaging results and quantitative analyses of morphology on the mycolic acid biosynthetic pathway, we noted highly analogous morphotypes irrespective of the gene targeted for silencing (*Figure 9C and D*). It was also evident that knockdown of components involved in early steps produced similar results to inhibition of the final step, involving the flippase, MmpL3 (*Xu et al., 2017*; *Su et al., 2019*). Given the consistency observed following genetic disruption of mycolic acid synthesis, we wondered whether chemical inhibition would produce a comparable morphological change. To test this possibility, we treated cells with isoniazid (INH), a frontline anti-TB drug which inhibits the essential enoyl-ACP reductase, InhA (*Timmins and Deretic, 2006*; *Vilchèze and Jacobs, 2007*). Exposure to 1X minimum inhibitory concentration (MIC$_{90}$) INH triggered increases in cell width and morphological alterations which were closely similar to those produced by genetic silencing of *inhA* (*Figure 9C and D*). This was an important observation since it suggested the possibility that, by analogy with bacterial cytological profiling (*Nonejuie et al., 2013*; *Nonejuie et al., 2016*), drug-induced morphological changes could be utilized to inform antimycobacterial MOA more broadly; that is, by exploiting the CRISPRi-imaging database of correlated genotype-phenotype connections with chemical (drug-induced) phenotype.

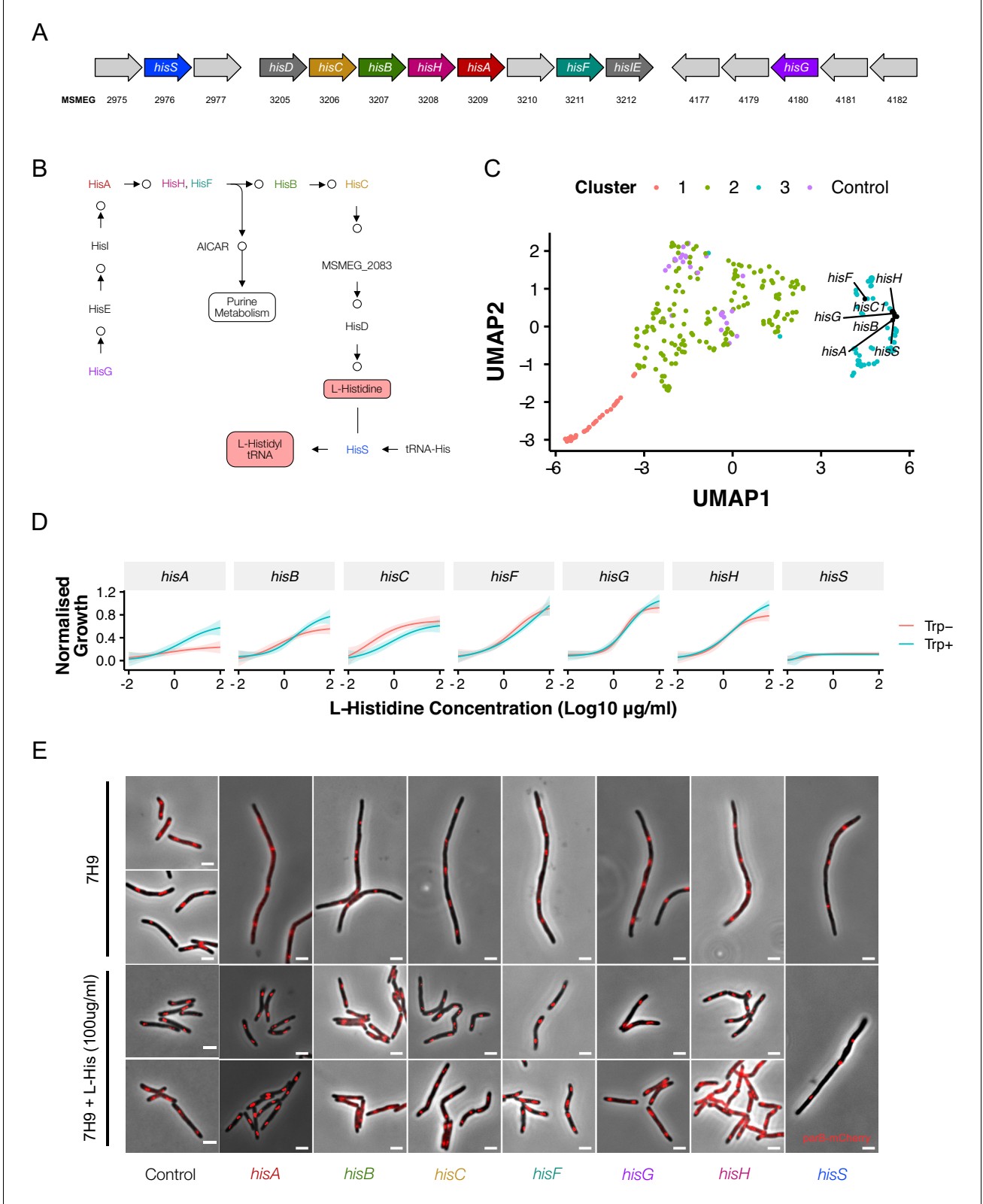

**Figure 8.** Disrupting components of histidine biosynthesis produces a filamentation response.  (A) Histidine biosynthetic genes are located throughout the chromosome. (B) Histidine is synthesized through a multi-step pathway (KEGG). (C) Cluster analysis demonstrated a consistent filamentation phenotype associated with genes involved in histidine synthesis. (D) Growth could be rescued with histidine supplementation (normalized to non-
*Figure 8 continued on next page*

*Figure 8 continued*

induced mutants). As HisA has an additional role in tryptophan synthesis, full rescue required simultaneous tryptophan supplementation (50 µg/ml). (E) The addition of histidine to ATc-containing growth media rescued the filamentation phenotype in all mutants other than hisS.

The online version of this article includes the following figure supplement(s) for figure 8:

**Figure supplement 1.** Absence of septum formation on histidine depletion.

## CRISPRi phenoprinting for antimycobacterial MOA determination

To ascertain if the CRISPRi dataset could be used to determine antimycobacterial MOA, we investigated a panel of compounds from different classes that inhibit various essential mycobacterial processes (*Figure 10A*). For proof-of-concept assays, the initial selection was limited to five clinically used anti-TB drugs: the first-line agents, INH and ethambutol (EMB), which disrupt cell-wall biosynthesis (*Abrahams and Besra, 2018*); RIF, which binds the DNA-dependent RNA polymerase subunit, RpoB (*Koch et al., 2014*); the second-line fluoroquinolone, moxifloxacin (MOXI), which inhibits DNA gyrase (*Kumar et al., 2014*) and bedaquiline (BDQ), an inhibitor of mycobacterial ATP biosynthesis recently approved for the treatment of MDR-TB (*Sarathy et al., 2019*). Thereafter, the panel was expanded to include an additional 12 known and experimental antimycobacterial compounds (*Figure 10—figure supplements 1–3*).

Morphotypes observed following drug treatment were often positioned in regions of UMAP space consistent with their known MOAs. For example, cells exposed to BDQ at 2X and 4X MIC clustered tightly with components of the mycobacterial ATP synthase and mapped to a region enriched for genes involved in energy metabolism (*Figure 10A*); the nearest neighbor for both 2X and 4X BDQ-treated cells was *atpA* (*Figure 10B*). The cell-wall targeting compounds similarly produced closely correlating profiles: EMB-treated cells clustered tightly together, irrespective of applied concentration, and were positioned in lipid metabolism-enriched UMAP space. The nearest neighbor at 2X and 4X MIC was *pks16*, a gene involved in mycolic acid synthesis; moreover, consistent with inhibition of arabinogalactan synthesis, *aftA*, *ubiA*, *dprE1*, *glfT2* were proximal to the EMB-exposed cells at all three applied drug concentrations. INH-treated cells were situated in lipid metabolism-enriched space at 1X MIC but, at higher concentrations, fell in a region associated with energy production and conversion. Notably, however, the nearest neighbor at 4X MIC was *fabD*, a component of mycolic acid synthesis. In contrast, compounds targeting peptidoglycan, including vancomycin and D-cycloserine, did not yield definitive profiles (*Figure 10—figure supplements 1–3*). Manual inspection of the data indicated that this was probably due to the induction of lysis, highlighting a limitation inherent in applying a single, 18 hr endpoint for analysis of all drug treatments.

Compounds targeting DNA replication generally produced filamentation, and clustered accordingly. Notably, MOXI-treated cells did not co-localize with *gyrA* and *gyrB* knockdowns (the DNA-gyrase subunits, and targets of MOXI) in UMAP space and were instead associated with DNA metabolism, an observation common to all fluoroquinolones. In contrast, novobiocin (NVB), a gyrase B inhibitor (*Chatterji et al., 2001*), did not trigger filamentation, instead positioning closely with *gyrA* and *gyrB* genetic knockdowns. Like the fluoroquinolones, the experimental agents nargenicin (putative DnaE1 inhibitor; *Painter et al., 2015*) and griselimycin (which binds to the β-clamp, DnaN; *Kling et al., 2015*) were associated with DNA metabolism pathways, in this case mapping much closer to their known or predicted targets.

While compounds disrupting DNA replication and cell envelope biogenesis were readily associated with a target class, this was not universally true for drugs with other MOAs. At 4X MIC, RIF was associated with *rpoB*. However, at the two lower concentrations, RIF-treated cells were positioned at dramatically different locations in UMAP space.

It was noticeable, too, that compounds targeting mycobacterial protein synthesis – for example, linezolid (*Leach et al., 2011*), streptomycin or kanamycin (*Riska et al., 2000*) – similarly failed to produce responses analogous to their cognate gene knockdowns. Like RNA polymerase, the protein translation machinery operates as a large, multi-protein complex, perhaps exposing a deficiency in this approach when applied to the macromolecular machines responsible for transcription and translation. Overall, however, the results supported the utility of CRISPRi-enabled cytological profiling to inform compound MOA, especially for agents targeting DNA replication, cell-wall biosynthesis, and energy metabolism.

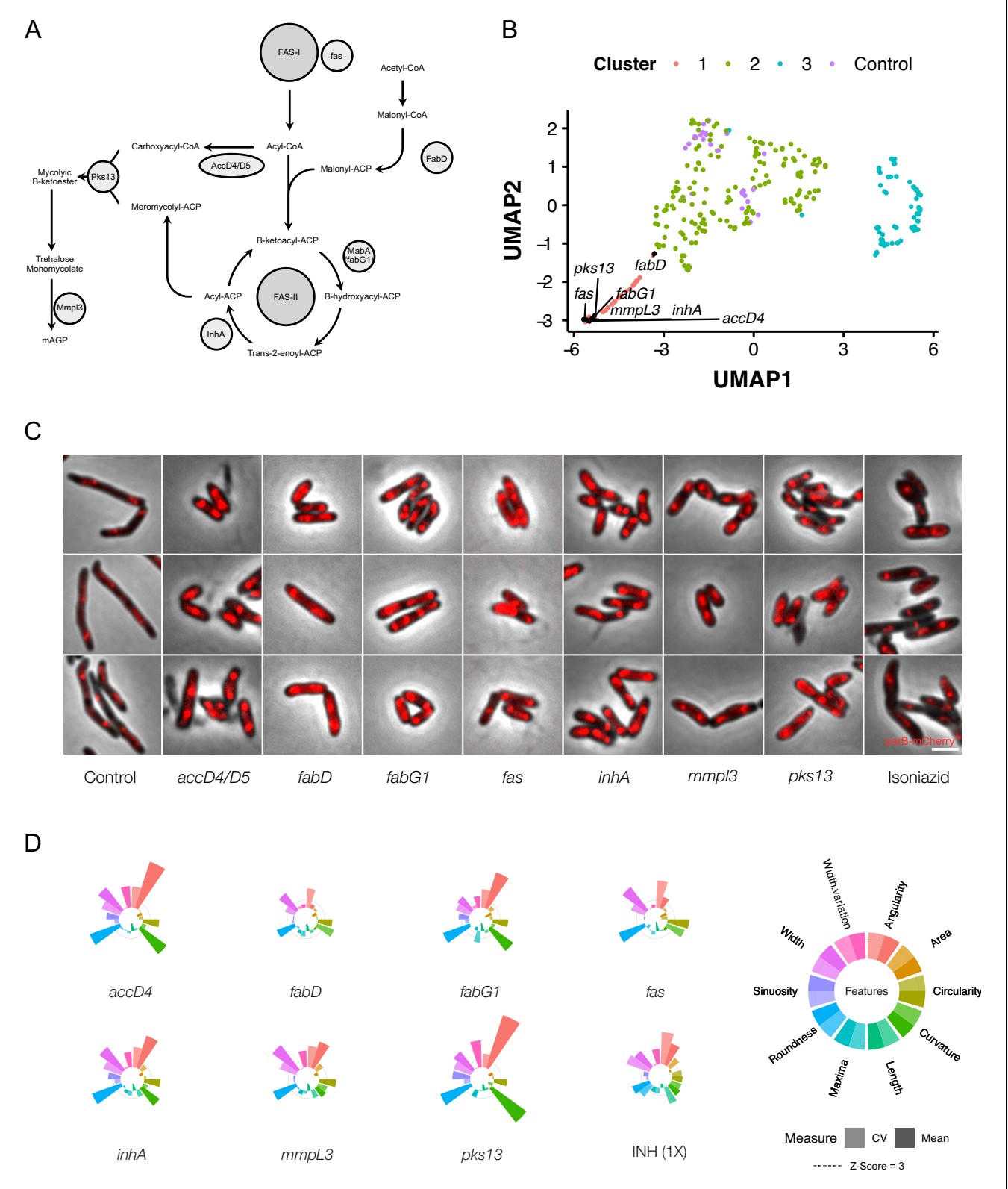

**Figure 9.** Genetic or chemical inhibition of mycolic acid biosynthesis produces a distinctive change in bacillary morphology. (**A**) The multi-step pathway of mycolic acid biosynthesis. (**B**) Components of mycolic acid synthesis, visualized in UMAP space, cluster closely and are morphologically similar (**C**) with overlapping phenoprints (**D**). Exposure to the frontline anti-TB drug, isoniazid (INH), at 1X MIC produces a similar morphological change and an analogous phenoprint to silencing of inhA, the validated target of INH (*Vilchèze and Jacobs, 2007*).

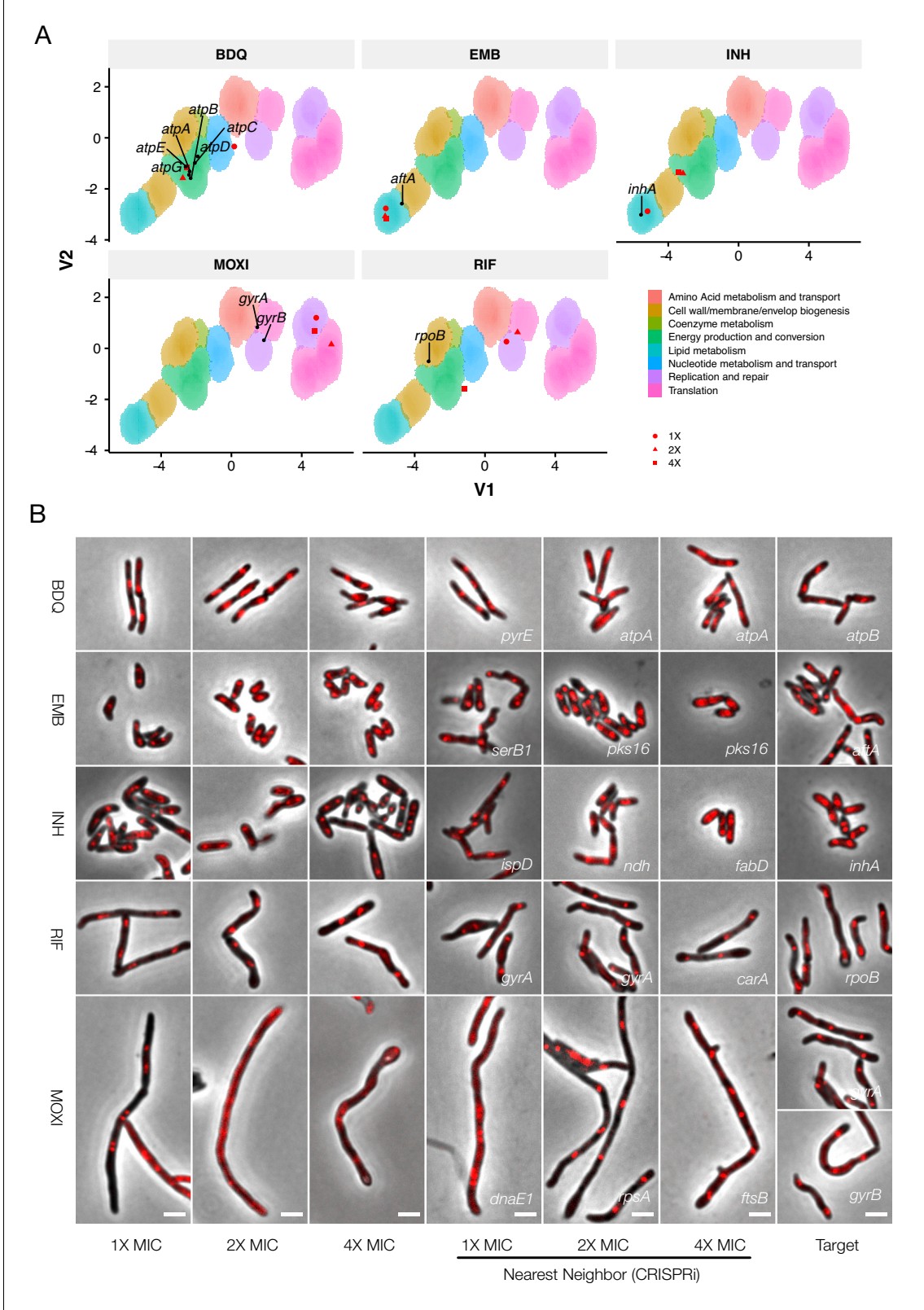

**Figure 10.** Phenoprinting can inform antimycobacterial MOA. Cells were exposed to varying supra-MIC concentrations (1X, 2X, 4X MIC) of the selected antimycobacterial compounds for 18 hr, imaged, and analyzed using the pipeline developed for CRISPRi-imaging. The resulting profiles were visualized in CRISPRi-generated UMAP space (**A**), and the morphological nearest-neighbor identified (**B**). Known targets were simultaneously visualized for comparative purposes. BDQ, bedaquiline; EMB, ethambutol; INH, isoniazid; RIF, rifampicin; MOXI, moxifloxacin.

*Figure 10 continued on next page*

*Figure 10 continued*

The online version of this article includes the following figure supplement(s) for figure 10:

**Figure supplement 1.** Phenoprinting can inform antimicrobial MOA.
**Figure supplement 2.** Phenoprinting can inform antimicrobial MOA.
**Figure supplement 3.** Phenoprinting can inform antimicrobial MOA.

## Discussion

We have described the construction and validation of an arrayed library of CRISPRi mutants targeting essential *M. smegmatis* homologs of *M. tuberculosis* genes. By coupling the library with a high-throughput imaging pipeline, we confirmed the utility of quantitative methodologies to describe morphological changes and, moreover, established the capacity to harness morphotypic information to construct a database of cytological phenotypes, or mycobacterial phenoprints. Then, in demonstrating the potential utility of the resulting phenoprint atlas, we presented four examples illustrating its use in enabling preliminary characterization of single-gene function (DNA methylase), the identification of distinct morphotypes associated with biosynthetic (histidine) and macromolecular (mycolic acid) pathway disruptions, and the potential to exploit CRISPRi morphotypes as a chemical-genetic tool to illuminate antimycobacterial MOA determination.

Advances in imaging technologies (*Camsund et al., 2020*) and sample preparation (*Shi et al., 2017*), together with the availability of software tools such as MicrobeJ (*Ducret et al., 2016*), have enabled the application of imaging in bacterial systems biology (*Huang, 2015*). Our pipeline was optimized for medium-throughput imaging of mycobacteria; however, increasing the throughput of imaging, for example with the Strain Library Imaging Protocol (*Shi et al., 2017*), would be advantageous for larger collections of mutants. To ensure completeness in our online 'Morphotypic Landscape' database, we plan in future to expand the library to include all 423 predicted essential *M. smegmatis* genes from our pooled screening data (*de Wet et al., 2018*). There appears to be significant value, too, in establishing an equivalent library in *M. tuberculosis*, a challenging undertaking but one which would benefit from the lessons learnt from this study. In this context, we note with interest key recent advances in mycobacterial cytological profiling (*Smith et al., 2020*) which suggest the potential to combine these complementary approaches in applying CRISPRi-imaging to *M. tuberculosis*.

In developing our analytical pipeline, we relied heavily on the pioneering work of Jacobs-Wagner and colleagues (*Campos et al., 2018*). In particular, we adapted a machine learning-based cleanup that leverages the power of MicrobeJ but adds a *Mycobacterium*-specific post-processing step. This classifier should have downstream utility for future imaging studies in *M. smegmatis* but will require retraining if utilized for *M. tuberculosis*. For analysis, we applied an unsupervised learning approach that combined UMAP dimensionality reduction with density-based clustering of our multidimensional dataset. UMAP is a generally applicable algorithm, but has enjoyed especially rapid uptake for the visualization of single-cell sequencing results (*Becht et al., 2018*). In our pipeline, we also used UMAP for data visualization and as a preprocessing step prior to the application of density-based clustering approaches, analogous to the use of t-SNE by Jacobs-Wagner and colleagues (*Campos et al., 2018*). While non-linear dimensionality reduction techniques must be used with caution, our extensive validation supports the utility of the approach. Additionally, our dataset should be amenable to further analysis using alternative analytic approaches, including deep-learning (*von Chamier et al., 2019*).

Using data obtained from our CRISPRi screen, we generated a quantitative atlas of morphological changes associated with essential gene silencing in *M. smegmatis*. For ~90% of the strains in our library, no prior morphological data were available at the outset of the study. Moreover, approximately 40% of the genes and their protein products had not been characterized at all, either functionally (biochemically or microbiologically) or structurally. It was notable, therefore, that almost 78% of the essential genes assayed here resulted in at least one dramatically aberrant morphological feature on CRISPRi knockdown. This number exceeds the ~60% of *B. subtilis* mutants that produce clear terminal phenotypes (*Peters et al., 2016*) and may reflect increased sensitivity of our analytic approach and/or the differential compositions of the respective libraries. However, given that only 20% of the non-essential *E. coli* Keio Collection displays significant morphological changes

(*Campos et al., 2018*), it is evident that silencing of essential genes is more likely to disrupt core processes affecting maintenance of general bacillary morphology.

A number of CRISPRi mutants produced morphological alterations that were unexpected. For example, while filamentation is a well-described phenomenon consequent on interference in cell-division (*Dziadek et al., 2003*; *Campos et al., 2018*; *Wu et al., 2018*), DNA replication (*Greendyke et al., 2002*; *Justice et al., 2008*) and depletion of ClpP protease components (*Li et al., 2010*), it is not typically associated with genes involved in protein synthesis or export. Nevertheless, we recorded filamentation responses following silencing of ribosomal subunits, the Sec translocase, and histidine biosynthesis. These phenotypes contrast with *B. subtilis* where filamentation is not observed on silencing of the corresponding genes (*Peters et al., 2016*). In *Salmonella typhimurium* (*Murray and Hartman, 1972*) and *E. coli* (*Frandsen and D'Ari, 1993*), mutants overexpressing *hisH* and *hisF* are known to produce filaments, possibly through a scarcity of PBP3 substrates required for cell division (*Cano et al., 1998*). To our knowledge, though, filamentation of histidine auxotrophs has not been reported in any bacterial species, including mycobacteria. The mechanism underlying the filamentation phenotype is unclear, though potentially attributable to interference in cell division through an unknown mechanism. Moreover, the fact that this phenotype manifests on knockdown of the L-histidine-tRNA ligase, *hisS*, suggests the involvement of a regulatory mechanism dependent on the presence of his-tRNAs (*Raina and Ibba, 2014*). However, further work is required to explain this observation.

While interfering in cell envelope biogenesis might be expected to cause aberrations in morphology, the increases in mycobacterial cell width observed on silencing essential steps in mycolic acid synthesis have limited precedent in the literature. One example is the association of temperature-sensitive mutants of *inhA* with analogous phenotypes (*Vilchèze et al., 2000*). The phenotypic consistency across the pathway was nevertheless surprising; again, however, the precise mechanism remains elusive. It is possible that the cell-wall synthetic machinery acts together in a spatiotemporally related complex that requires the presence of all proteins for function. It might be instructive, for example, that fluorescently-tagged versions of InhA and MabA co-localize (*Vilchèze et al., 2000*) and may be present in the same specialized membrane domain (*Hayashi et al., 2018*). Knockdown of individual components of the pathway might therefore impact multiple protein–protein interactions, collapsing numerous components of cell-wall synthesis to produce the same phenotypic outcome. Alternatively, it is possible that biosynthetic precursors accumulate, producing consistent alterations in cell structure. A further possibility is that mycolic acids might play a more important structural role in the maintenance of mycobacterial cell width – an intriguing prospect considering the absence of the prokaryotic actin homolog, MreB, in mycobacteria (*Singh et al., 2010*). Further work is needed to resolve these possibilities and should benefit from the collection of CRISPRi mutants described here.

In general, the CRISPRi-induced morphological changes appeared consistent within pathways and following exposure to the various anti-mycobacterial drugs. This was especially true of genes involved in DNA replication, cell division, cell-wall metabolism, and energy metabolism – a property which allowed us to leverage the platform to assign functional predictions for a number of genes, and to infer antimicrobial MOA. An example is MSMEG_3213, which morphological profiling, transcriptomic profiling and combinatorial CRISPRi identified as methylase of a predicted Type II R-M system in *M. smegmatis*. The biological function of this system is currently unclear; however, one practical implication is that, if involved as expected in intrinsic phage defense, it might inadvertently eliminate phages, which could be otherwise useful for clinically relevant mycobacteria such as *M. tuberculosis* or *M. abscessus* (*Dedrick et al., 2019*). It is tempting to consider, too, if the differential complement of R-M systems among mycobacteria might determine the apparent species-selectivity of some mycobacteriophages – for example, DS6A (*Mayer et al., 2016*) – and whether the MSMEG_3213 methylase, which is conserved in *M. tuberculosis*, functions in cell-cycle regulation (*Wion and Casadesús, 2006*) in addition to its role in self-defense.

Cytological profiling has been extensively exploited to determine antimicrobial MOA (*Nonejuie et al., 2013*; *Huang, 2015*; *Nonejuie et al., 2016*), with a very recent report providing the first evidence of its utility in mycobacteria (*Smith et al., 2020*). To our knowledge, all approaches have consistently utilized whole-cell drug treatment to create profiles of known mechanisms. In this work, we instead applied a genetic approach, generating a database of phenoprints based on essential gene knockdown which were then compared with profiles produced by drug exposure of whole

cells. Notwithstanding the fundamental difference between genetic (transcriptional) knockdown and pharmacological (small molecule) inhibition, we observed strong overlap in the resulting phenotypes, especially for compounds targeting energy metabolism (BDQ), the mycolic acid (INH) or arabinoga-lactan (EMB) components of the cell wall, as well as DNA metabolism (fluoroquinolones and griseli-mycin). In some cases, we noted, too, that different concentrations of compound could produce disparate morphological profiles, suggesting a possible dependence of MOA on drug concentration (*Bernier and Surette, 2013*). Furthermore, not all compounds produced defining phenotypes. For example, compounds targeting peptidoglycan synthesis (DCS and VAN) led to rapid lysis, while ami-noglycosides did not cluster with the majority of ribosomal knockdowns. While these exceptions expose limitations in cytological profiling (discussed below), the approach nevertheless appears use-ful as rapid pre-screen of compound MOA in drug discovery pipelines, prior to more in-depth inves-tigation. Moreover, the pioneering development by Aldridge and colleagues (*Smith et al., 2020*) of the MorphEUS (morphological evaluation and understanding of stress) system for mycobacterial cytological profiling suggests the potential to replicate the CRISPRi approach described here in *M. tuberculosis*, a daunting but tractable challenge.

It was notable that MOXI-treated cells did not sit closely with *gyrA* and *gyrB* knockdowns and were instead associated with DNA metabolism, an observation common to other fluoroquinolones. While this separation in UMAP space might be deemed surprising, two factors are salient to the interpretation of this result: firstly, the role of mycobacterial DNA gyrase in the removal of torsional stress and the maintenance of template topology during RNA transcription (*Ahmed et al., 2017*) is consistent with the dominant effect of *gyrA* and *gyrB* knockdown manifesting as more similar to RNA polymerase inhibition (here, the close association with *rpoB* in the UMAP analysis is potentially telling); and, secondly, the filamentation observed on fluoroquinolone exposure is SOS-dependent (*Drlica et al., 2008*) and involves the formation of lethal double-strand DNA breaks following drug-mediated trapping of the DNA-gyrase complex – an effect which ATc-induced CRISPRi knockdown does not produce. Supporting this interpretation, the gyrase B inhibitor, NVB, did not trigger fila-mentation, and was instead closely associated with *gyrA* and *gyrB* knockdowns, a key observation given previous work noting the lack of SOS induction in NVB-treated cells (*Boshoff et al., 2004*). A question which commonly arises is how the apparently discrepant *gyrA*/*gyrB* and moxifloxacin phe-noprints can be reconciled with recent data from a pioneering high-throughput chemical-genetic screen which reported the identification of novel gyrase inhibitors by utilizing a library which included *gyrA* and *gyrB* hypomorphs (*Johnson et al., 2019*). In addressing this dilemma, it is impor-tant to remember that hypomorphs provide a powerful means to identify hypersusceptibilities con-sequent on bacillary exposure to chemicals which inhibit the same (cognate) gene or pathway targeted by the knockdown system (i.e. the genetic and antibiotic targets are identical); in contrast, whole-cell morphotypes report on the physiological response to (or manifestation of) lethal stress induced by essential gene silencing. Importantly, a key element distinguishing antibiotic-mediated target inhibition or corruption from transcript depletion is time: antibiotic treatment inhibits (or cor-rupts) active processes, triggering intracellular catastrophe (in the case of MOXI treatment, an SOS response), whereas gene silencing is much more gradual, or ordered, avoiding the impact of instan-taneous loss of function.

As for all large-scale assays, the approach detailed here inevitably carries inherent limitations which must be borne in mind when analyzing and applying the data: (i) Although motivated primarily by pragmatism, the decision to apply an 18 hr endpoint for all analyses might inadvertently have biased the 'strongest' phenotypes to genes involved in DNA replication, cell division, and cell-wall metabolism. One obvious complication pertains to the drug MOA analyses, and is evident in the striking frequency with which the data points for 1X, 2X and 4X concentrations map differently in UMAP space for different compounds. Intuitively, increasing drug concentration might be expected to have two principal effects – increasing the rate of cell death while retaining the primary MOA (put simply, cells treated at 4X MIC will be further along the 'path to death' at 18 hr than those treated at 1X MIC) or increasing the likelihood of hitting secondary targets. This issue has been noted previ-ously in transcriptional profiling of the *M. tuberculosis* drug-exposure response (*Boshoff et al., 2004*) in which optimizing the drug concentration and exposure time were critical in avoiding con-vergence on a common stress pathway, obscuring the informative, drug-specific transcriptional sig-nature. Similarly, the propensity for the rate of gene-product depletion to impact qualitatively the terminal phenotype has been elegantly demonstrated in *B. subtilis* (*Peters et al., 2016*).

Notwithstanding the significant resources required, future work might therefore envisage the use of time-lapse imaging of all mutants to identify optimal gene-specific knockdown durations. (ii) This proof-of-concept study employed only aerobically grown mycobacteria cultivated at 37°C in media comprising Middlebrook 7H9 broth base, with glycerol as primary carbon source. There is ample precedent in the literature indicating that gene essentialities, metabolic vulnerabilities and drug susceptibilities can alter as a function of growth conditions; therefore, the phenotypes presented here are necessarily condition-specific, and could benefit significantly from expanding these analyses to other, 'disease-relevant' culture systems. By extension, there is also the potential to perform equivalent analyses in different genetic backgrounds, avoiding the well-described pitfalls of 'wild-type' laboratory strains. (iii) Morphological profiling appears to offer a rapid means of preliminary gene function assignment or compound MOA; however, definitive validation is required – via further biochemical and/or functional analysis – to ratify the functional assignments predicted using this tool. (iv) Future iterations should also aim to resolve the large cluster (Cluster 2) of genes exhibiting minimal morphological effects on knockdown into more informative sub-clusters, perhaps through the incorporation of additional features and/or by applying alternative fluorescent reporters (possibly in addition to the ParB reporter). Some approaches to consider include the use of sentinel genes and transcription factors involved in different aspects of macromolecular biosynthesis and/or metabolic remodeling (*Naran et al., 2016*; *Boot et al., 2018*). In addition to the potential for improved MOA delineation, such refinements are necessary to address the large number of hypothetical proteins whose functions were not significantly informed by the current morphological profiling algorithm.

In conclusion, we are hopeful that the library of *M. smegmatis* mutants – and the associated online database – will offer a potentially valuable resource for the mycobacterial research community, particularly given the utilization of the ParB-mCherry reporter background which, to our knowledge, is unique in coupling whole-cell morphological information with a measure of replicative status and ploidy. While a large collection of inducible protein-degradation mutants was recently described in *M. tuberculosis* (*Johnson et al., 2019*), there is not yet an equivalent in the faster-growing model mycobacterium, *M. smegmatis*. Moreover, in addition to the validated strains, the plasmids used to establish our library should enable relatively simple construction of either single mutants or mutant libraries in alternative genetic backgrounds and reporter strains.

## Materials and methods

### Key resources table

| Reagent type (species) or resource | Designation | Source or reference | Identifiers | Additional information |
|---|---|---|---|---|
| Strain, strain background (*Eschericia coli*) | Dh5a | | | |
| Strain, strain background (*M. smegmatis*) | ParB-mCherry | *Santi and McKinney, 2015* | | |
| Strain, strain background (*M. smegmatis*) | See *Supplementary file 1* | | | |
| Recombinant DNA reagent | pJR962 | *Rock et al., 2017* | | |
| Recombinant DNA reagent | See *Supplementary file 1* | | | |
| Sequence-based reagent | See *Supplementary file 1* | | | |
| Commercial assay or kit | OneTaq Master Mix | NEB | NEB M0482L | |
| Commercial assay or kit | Esp31-FastDigest | Thermo | FD0454 | |

*Continued on next page*

*Continued*

| Reagent type (species) or resource | Designation | Source or reference | Identifiers | Additional information |
|---|---|---|---|---|
| Commercial assay or kit | Zyppy-96 Plasmid Miniprep Kit | Zymo | D4041 | |
| Commercial assay or kit | FastRNA Blue Kit | MP Biomedicals | MP6025050 | |
| Commercial assay or kit | DNAse | NEB | NEB M0303S | |
| Commercial assay or kit | Poly(A) Polymerase | NEB | NEB M0276L | |
| Commercial assay or kit | Zymo RNA Clean and Concentrator-5 | Zymo | R1 013 | |
| Commercial assay or kit | Ribo-Zero rRNA Removal Kit | Illumina | MRZMB126 | |
| Commercial assay or kit | Nanopore Direct RNA Sequencing Kit | Oxford Nanopore Technologies | SQK-RNA002 | |
| Commercial assay or kit | Nanopore Direct cDNA Sequencing Kit | Oxford Nanopore Technologies | SQK-DCS109 | |
| Commercial assay or kit | NEBNext Q5 High-Fidelity Polymerase Master Mix | NEB | NEB M0543L | |
| Commercial assay or kit | QIAquick PCR purification kit | Qiagen | 28106 | |
| Commercial assay or kit | T4 Ligase | NEB | NEB M0202M | |
| Commercial assay or kit | SapI | NEB | NEB R0569S | |
| Chemical compound, drug | See *Supplementary file 1* | | | |
| Software algorithm | FIJI | *Schindelin et al., 2012* | | |
| Software algorithm | MicrobeJ | *Ducret et al., 2016* | | |
| Software algorithm | R | | | |
| Software algorithm | UMAP | *Lel et al., 2018* | | |
| Software algorithm | hdbscan | *Lel et al., 2017* | | |
| Software algorithm | tSNE | *Maaten and Hinton, 2008* | | |
| Software algorithm | Minimap2 | *Li, 2018* | | |
| Software algorithm | samtools | *Li et al., 2009* | | |
| Software algorithm | deepTools2 | *Ramírez et al., 2016* | | |
| Software algorithm | featureCounts | *Liao et al., 2014* | | |
| Software algorithm | DESeq2 | *Love et al., 2014* | | |
| Software algorithm | Iris | *Kritikos et al., 2017* | | |

## Construction of arrayed library

We limited our initial selection to 294 genes. In addition to the 288 genes found universally essential in both Tn-seq and CRISPRi-Seq (*de Wet et al., 2018*), we added six components of the ATP synthase which were found to be non-essential in Tn-seq owing to gene duplication in *M. smegmatis* (*Dragset et al., 2019*).

While CRISPRi is a reliable technology for producing knockdown, it is subject to variation in knockdown efficacy, depending on the selected guide RNA. Moreover, when targeting a gene with CRISPRi, the efficacy of a particular guide is not known de novo, and activity prediction algorithms are still in their infancy, particularly for bacteria (*Wang et al., 2017*). As a result, it is generally best practice to target a gene with more than one sgRNA to account for differences in activity. However, in the case of arrayed mutant collections, this requirement becomes unwieldy; as such, previous arrayed CRISPRi mutant collections in bacteria have utilized one predicted sgRNA per gene (*Liu et al., 2017*; *Wang et al., 2017*). Our previous CRISPRi-Seq data (*de Wet et al., 2018*) allowed us to identify essential genes that were growth-suppressed by multiple sgRNAs, but additionally provided empirical data on sgRNA knockdown efficiency. As a result, we were able to identify the most efficient sgRNAs for any gene.

For each of the 294 selected genes, two oligonucleotides (*Supplementary file 1*) were synthesized at 50 nmol and annealed by LGC Biosciences. For five genes, an additional set of oligonucleotides was used to validate the observed phenotypes (*Figure 2—figure supplement 2*). Oligonucleotides were delivered in 96-well plates, and were resuspended in water to a final concentration of 100 µM. Cloning was performed as previously described (*Rock et al., 2017*) but adapted for scale. Briefly, pJR962 (*Rock et al., 2017*) was digested overnight with BsmBI-FastDigest (Thermo) and gel purified. Ligations were performed in 96-well plates with 1U T4 Ligase (NEB) and incubated at room temperate overnight. Ligation reactions were transformed into 5 µl High-Efficiency DH5α cells (NEB) with heat-shock, rescued in TY Broth, and plated on LB agar prepared in 6-well plates. Following overnight growth at 37°C, single colonies were picked into 800 µl of LB broth, in 2 ml deep 96-well plates, and grown overnight at 37°C with vigorous shaking. All *E. coli* cultures were supplemented with Kanamycin (Roche) at a final concentration of 50 µg/ml. Plasmids were extracted with a Zyppy-96 Plasmid Miniprep kit (Zymo) according to manufacturer instructions and quantified via Nanodrop.

Approximately 100 ng of plasmid was electroporated into the *M. smegmatis* ParB-mCherry reporter strain (*Santi and McKinney, 2015*) and rescued in 7H9 broth supplemented with OADC and 0.05% Tween 80, for 3 hr at 37°C. Electroporation reactions were plated on 7H10 agar supplemented with OADC and Kanamycin at 20 µg/ml, prepared in 6-well plates, and grown at 37°C until visible colonies formed. Single colonies were picked into 400 µl 7H9 supplemented with kanamycin (20 µg/ml) and grown with shaking until stationary phase. Glycerol stocks of all transformants were stored at −80°C. Any failed ligations or transformations were repeated.

## Sequencing validation

PCR Primers (*Supplementary file 1*) designed to target the sgRNA-containing region of plasmid pJR962 were synthesized by Inqaba Biotech. PCR reactions were prepared with OneTaq Master-mix (NEB) in 96-well plates to a final volume of 20 µl. Using pipette tips, scrapings of glycerol stocks were taken and mixed with the PCR reaction. The PCR was performed with the settings described (*Supplementary file 4*). PCR reactions were purified and sequenced by the Stellenbosch Central Analytic Facility using the same forward primer as the PCR reaction. Sequencing results were analyzed using a custom written R script. For PCR amplification, we opted not to use a high-fidelity polymerase owing to the number of reactions performed. Since this risked introducing artificial mismatches or ambiguities into the sequencing results, a sequence was considered correct if the mutant had a high-quality alignment with its expected sequence. We accepted ambiguities in the sequencing trace if the strain functionally validated in downstream growth analysis. PCR reactions that did not produce sequencing results were repeated; ultimately, sequence-validation was obtained for 238 mutants from the initial cloning workflow, a success rate of 83%. Of the invalidated mutants, 15 wells contained plasmid backbone, 15 were cross-contaminants which occurred at some point during the cloning process, and the remaining 20 failed to sequence. We maintained the plasmid-backbone strains in our library and downstream workflow as these represented useful empty

vector sequences, so that control strains were nested within the arrayed library. Cloning was repeated for invalidated mutants.

## Growth validations

Square plates containing standard Middlebrook 7H10 OADC agar were prepared and supplemented with kanamycin (Roche,20ug/ml), with or without anhydrotetracycline (ATc, Sigma, 100 ng/ml). Fresh cultures were inoculated from glycerol freezer stocks into 400 μl Middlebrook 7H9 liquid medium supplemented with kanamycin (Roche, 20 μg/ml) in deep 96-well plates. Cultures were grown with shaking until saturated. Stationary-phase cultures were diluted 1:10 000 into fresh 7H9 before spotting onto prepared 7H10 plates using a 96-well replicator pin (Sigma). All spotting was performed in triplicate. Plates were grown until colonies formed and photographed using a lightbox. Colony sizes were quantified using Iris (*Kritikos et al., 2017*).

## Image sample preparation

Fresh cultures were inoculated from glycerol stocks into 400 μl 7H9 with kanamycin (Roche, 20 μg/ml) and grown until saturated. Saturated cultures were diluted 1:800 into fresh 7H9 with kanamycin (Roche, 20 μg/ml) and grown for 24 hr until exponential phase. Exponential-phase cultures were inoculated 1:40 into 7H9 supplemented with kanamycin (Roche, 20 μg/ml) and ATc (Sigma 100 ng/ml) and grown for 18 hr, with shaking. Large-format agarose pads were prepared with water to a final concentration of 2%. Briefly, 2 ml of molten low-melt agarose was sandwiched between two rectangular coverslips (No. 1.5, 24 x 60 mm). The bottom coverslip was marked using a laser-cut stencil to indicate sample placement. Agarose pads were left to dry, and 1 μl of induced culture spotted onto the pads, using a multichannel pipette. Each pad contained eight samples. Twenty-four strains were imaged per day, and replicate imaging was performed for 137 samples to validate the reproducibility of the imaging workflow.

## Microscopy

Large-format agarose pads were imaged using a Zeiss Axio Observer Z1 and ZEN 2 (blue edition) software, with the ZEN Tiles and Positions and ZEN Autofocus Modules installed. Images were captured using a Zeiss Axiocam 503 with 3X analogue gain. Using a low magnification 10X objective, each sample was localized on the pad. Approximately 24 fields-of-view were selected with a 100X Phase Contrast Objective (1.4NA), before capturing images using bright-field and fluorescent imaging. Fluorescence was excited using a Colibri Green LED (555/30 nm) and filtered at 590–650 nm. Exposure times were maintained across imaging sessions for Brightfield Images, and for fluorescent images. A software autofocus regimen was used before each field-of-view was captured. In the case of low-density samples, fields-of-view were manually chosen and increased in number. Raw images were saved as CZI files prior to processing and data extraction.

Time-lapse imaging was performed on 1.5% agarose pads embedded with 7H9 OADC medium containing ATc (Sigma, 100 ng/ml) and kanamycin (Roche, 20 μg/ml), in glass-bottomed dishes (NEST Biotechnology). Cells were maintained in an incubated chamber at 37°C and imaged every 15 min with a software autofocus regimen implemented between each frame.

## Image processing and data extraction

All image processing was performed in FIJI (*Schindelin et al., 2012*). CZI Images were converted to TIFF images and were manually inspected and out-of-focus fields removed. To enhance foci in the fluorescent channel, a Gaussian blur filter was applied to the fluorescent channel and subtracted from the original image. Furthermore, overall intensity of each fluorescent channel was normalized throughout the imaging dataset, to a mean pixel intensity of 15. ImageJ processing scripts are available at https://osf.io/pdcw2/. Pre-processed images were analyzed using MicrobeJ (*Ducret et al., 2016*) with limited constraints on cell and foci detection. MicrobeJ output (cell contours, shape descriptions and fluorescent localization – see *Figure 2—figure supplement 1*) was exported as CSV files.

Quantitation of microcolony growth rates was performed in ImageJ and R with a custom script. Briefly, images were thresholded in ImageJ and the microcolony area extracted per frame. Time-lapse plots were produced in R using a combination of ggplot2 (*Wickham, 2016*) and gganimate.

## Dataset curation

To curate the output of MicrobeJ, a classifier was built in R. Two fields-of-view were sampled from each mutant and replica dataset and processed and analyzed as described above. All detected objects were initially exported. To create a reference of correctly identified cells, cells were inspected with the interface built into MicrobeJ, and misidentified cells removed from the dataset. Thus, two datasets were exported: the full dataset of all identified objects, and the dataset of classified cells. In total, approximately 22,000 objects were manually classified. The classified data were imported into R and utilized to build a classification model. Briefly, 20% of the dataset was reserved as a test sample, and the remaining 80% was used to train a variety of models, using the Caret package (*Kuhn, 2008*). For model training, fivefold cross-validation was used, and resampling performed across the default tuning parameters. ROC was used to select the optimal tuning parameters for each model. A selection of models was tested and compared based on ROC AUC. An Averaged Neural Network was empirically chosen as the best-performing model (*Figure 2—figure supplement 3*). The trained classifier was used to classify the entire dataset and remove misidentified objects. Data were input through the curation pipeline and labeled based on sequencing results and growth validations. Reproducibility of the imaging workflow was tested by comparing length of the replica image sets. The scripts used to develop and test the models, and all input data, are available at https://osf.io/pdcw2/.

## Representative cell generation

For each imaged mutant strain, we derived the mean values for a selection of 13 morphological features – Angularity, Area, Aspect Ratio, Circularity, Curvature, Feret, Length, Perimeter, Sinuosity, Solidity, Roundness, Width and Width Variation – and counts of ParB maxima. We utilized the approximate nearest neighbor approach in the R package, RANN, to identify a single cell that best approximated the mean values of the mutant. The cell contour, derived from MicrobeJ, was plotted using ggplot, and maxima identified by MicrobeJ were superimposed on the plotted contour. Visualizing scripts are available at https://osf.io/pdcw2/.

## Data input and processing

For data processing, we adapted a previously described approach (*Campos et al., 2018*). For each mutant, we calculated the mean of each measured feature, and the CV (mean/standard deviation). We transformed each feature into a Z-score, relative to the distribution of means derived from the empty vector strains utilizing the equation:

$$Z = \left(F_i - mean\left(F_i^{WT}\right)\right)/sd\left(F_i^{WT}\right)$$

The normalized Z-Score therefore represents the number of standard deviations away from the mean of the wild-type distribution (*Figure 4—figure supplement 1*). All subsequent analyses utilized Z-Score adjusted data, unless otherwise described.

## Cluster of Orthologous Groups (COG) Annotations

Each gene in our dataset was assigned to a Cluster of Orthologous Groups using annotations obtained from eggNOG (*Huerta-Cepas et al., 2016*). We utilized both *M. smegmatis* and *M. tuberculosis* annotations to assign COGs to as many genes as possible. In addition, we manually annotated a number of genes with recently identified function (*Wu et al., 2018*).

## Phenoprint visualization

For phenoprint visualization, variables were chosen for their interpretability and potential biological relevance. Variables included angularity, area, circularity, curvature, length, sinuosity, roundness, width, width variation and number of ParB foci (maxima). For visualization, the mean and CV Z-scores for each variable were displayed in a circular coordinate system. Where visualizations represented groups of mutants (for example COGs) means of the mutant Z-scores were used for visualization. Phenoprint visualization code is available at https://osf.io/pdcw2/.

## COG enrichment analysis

For each feature, we utilized a cut-off of 3 standard deviations above or below the mean of the control distribution, and tested for enrichment of each COG category utilizing Fisher's Exact Test. We corrected for multiple testing using the Benjamini-Hochberg Procedure and utilized a significance cut-off of $p < 0.05$. For display of data, we opted to replicate prior work (*Campos et al., 2018*) in displaying all genes found above or below the threshold. COG categories that were enriched for a particular feature were highlighted to differentiate them from the background. We only displayed plots where at least one COG was found to be statistically enriched (*Figure 4—figure supplement 2*) Scripts are available at https://osf.io/pdcw2/.

## Dimensionality reduction and analysis

Dimensionality reduction allows visualization of multidimensional data in two-dimensional space. Utilizing our normalized Z-score data, we performed dimensionality reduction in R. We tested a number of approaches for dimensionality reduction, including a Principle Component Analysis (PCA), t-SNE (*Maaten and Hinton, 2008*) and UMAP (*Maaten and Hinton, 2008*) algorithms. We superimposed these dimensionality reduction techniques with a hierarchical density-based spatial clustering of applications with noise (hdbscan) (*Lel et al., 2017*). For each technique, we tested a variety of optimization parameters to produce distinct clusters of points, while still maintaining a consistent cluster of wild-type samples. On comparison, UMAP produced the best separated clusters, while maintaining a relatively uniform wild-type cluster. UMAP is characterized by a degree of stochasticity during initialization (*Lel et al., 2017*); therefore, we generated 100 maps which were each clustered with hdbscan (minPts = 12). Consistently present clusters across initializations were determined with hierarchical clustering.

To explain the features that assigned points to particular clusters, we superimposed a color-gradient based on the mutant Z-Scores for a chosen set of features. Additionally, we generated heatmaps describing mean Z-scores of chosen features, for each cluster. We performed COG enrichment on each cluster, utilizing Fisher's Exact Test, and adjusted for multiple testing using the Benjamini-Hochberg Procedure. For visualizing sub-clusters within the UMAP, we adapted an approach inspired by the previously described spatial analysis of functional enrichment (SAFE) approach (*Baryshnikova, 2016*; *Campos et al., 2018*), to identify regions of the UMAP output that were enriched for particular functions. Briefly, for each point on our 2-dimensional UMAP projection, we examined surrounding points, within a predefined radius (the 10th percentile of the distribution of pairwise distances between all points) and tested for enrichment of either COG or KEGG ontologies using a hypergeometric test. We combined enrichment testing with a k-nearest neighbor interpolation approach to identify spatial regions of our UMAP manifold that were enriched for particular functions.

## Antimicrobial profiling

For each drug, twofold serial dilutions were prepared in 400 µl of 7H9-OADC medium in deep 96-well plates. Exponential-phase *M. smegmatis* ParB-mCherry cells were inoculated into prepared media at a 1:40 dilution and cultured with shaking for 18 hr. Prior to imaging, ODs were measured and a dose-response curve fitted to the results with the DRC package in R (*Ritz et al., 2015*). The $MIC_{90}$ was derived from the fitted curve and defined as 1X MIC. We validated that the $MIC_{90}$ was consistent with results from a fluorescent resazurin based assay. To this end, 10 µl of resazurin (Sigma) was added to 50 µl of cell culture, incubated for one hour at 37°C and color change determined visually. For imaging, cultures were spotted on agarose pads, and imaged and analyzed as previously described.

## Cell cycle heatmaps

A population of cells at a single-timepoint contains a degree of temporal information, as the population contains cells at different stages of the cell cycle (*Campos et al., 2018*). To visualize movement of ParB through the cell cycle, we utilized extracted maxima-localization data and cell length from MicrobeJ and processed the data further utilizing a bespoke R script. For each mutant, we arranged cells by length, and produced consensus heatmaps of ParB localization relative to cell length using ggplot2. Scripts are available at https://osf.io/pdcw2/.

## Nanopore RNA-seq library preparation

Cultures of *M. smegmatis* were grown, with shaking, to exponential phase (OD$_{600}$ ~0.9–1.0) in 15 ml of 7H9 OADC at 37°C prior to RNA extraction. For CRISPRi knockdowns, the strain of interest was inoculated from an exponential-phase culture into 15 ml of 7H9-OADC supplemented with kanamycin (Roche, 20 µg/ml) and ATc (Sigma, 100 ng/ml) and cultured for 18 hr at 37°C with shaking. RNA was extracted using a FastRNA Blue Kit (MP Biomedicals) and ethanol-precipitated, according to manufacturer instructions. Extracted RNA was quantified by Nanodrop. Ten µg of RNA was treated with 2U DNase (NEB) at 37°C for 10 min, prior to cleanup with a Zymo RNA Clean and Concentrator-5 kit (Zymo Research) according to manufacturer instructions. The purified DNAse-treated RNA was poly-A tailed using *E. coli* Poly(A) Polymerase (NEB), according to manufacturer instructions, prior to clean-up with a Zymo RNA Clean and Concentrator-5 kit (Zymo Research). Ribosomal RNA was depleted with a Bacterial Ribo-Zero rRNA Removal Kit (Illumina) according to manufacturer instructions, prior to clean-up with a Zymo RNA Clean and Concentrator-5 kit (Zymo Research). Library preparation for Nanopore sequencing was performed using either a Nanopore Direct RNA Sequencing Kit (SQK-RNA001, Oxford Nanopore Technologies) or Nanopore Direct cDNA (SQK-DCS109, Oxford Nanopore Technologies) according to manufacturer protocols, and sequenced on MinION sequencers (Oxford Nanopore Technologies). Base-calling was performed by MinKNOW software (Oxford Nanopore Technologies), and data stored as FASTQ files for downstream analysis.

## RNA-Seq data analysis

Reads were aligned to the *M. smegmatis* mc$^2$155 genome (NC_008596) with Minimap2 (*Li, 2018*), utilizing the map-ont command. Produced SAM files were sorted and indexed with samtools (*Li et al., 2009*). For visualization, coverage was calculated using the bamCoverage command of deepTools2 (*Ramírez et al., 2016*) with normalization by Reads Per Kilobase per Million mapped reads (RPKM). featureCounts (*Liao et al., 2014*) was utilized for assigning read counts to genes, and DESeq2 (*Love et al., 2014*) was used for comparisons of upregulated transcripts. All visualizations were produced in R, using ggplot2.

## Dual CRISPRi Knockdown

sgRNAs were chosen to target *MSMEG_3214* and cloned into pJR962 as previously described (*Love et al., 2014*). Golden Gate (*Supplementary file 3*) primers were designed to amplify the promoter-sgRNA-terminator region of the plasmid, with SapI restriction sites included at the 5' region of the primers, and synthesized by Inqaba Biotech. The PCR reaction was performed with Q5 high-fidelity polymerase master mix (NEB) and the product column purified with a QIAquick PCR purification kit (Qiagen) and quantified by Nanodrop. PCR primers and reaction settings are available in *Supplementary file 4*. Golden Gate cloning was performed utilizing the purified MSMEG_3213-targeting plasmid cloned during establishing the library as a backbone. Briefly, a single-pot reaction was set up containing T4 ligase (10 000U) and buffer (NEB), SapI (NEB, 10Units), 75 ng of backbone, and purified PCR product at a molar ratio of 2:1. The reaction was incubated at 37°C for one hour, and at 55°C for 5 min, prior to transformation into *E. coli* DH5α (NEB). Colonies were screened by PCR using the sequencing primers, plJR962_Seq_F and plJR962_Seq_R, and OneTaq polymerase (NEB), as described previously, and positive clones sequenced by the Stellenbosch Central Analytic Facility utilizing plJR962_Seq_F to confirm the insertion. The sequence-verified plasmid was electroporated into *M. smegmatis* mc$^2$155 and tested for growth rescue by spotting assays.

## Acknowledgements

This work was supported by grants from the US National Institute of Child Health and Human Development (NICHD) U01HD085531 (to DFW), the Research Council of Norway (R and D Project 261669 'Reversing antimicrobial resistance') (to DFW), the South African Medical Research Council (SAMRC) (to VM), the National Research Foundation of South Africa (to VM and DFW), and core funding from the Wellcome Trust [203135/Z/16/Z]. Research reported in this publication was supported by the Strategic Health Innovation Partnerships (SHIP) Unit of the SAMRC with funds received from the South African Department of Science Innovation. TdW received partial funding support from the SAMRC through its Division of Research Capacity Development under the National Medical Students

Research Training Programme, from funding received from the South African National Treasury. We are thankful to Jeremy Rock and Sarah Fortune for the CRISPRi vectors, Keith Derbyshire and Todd Gray for the Dendra-tagged translational fusions, Bree Aldridge and Trever Smith for useful discussions, and Mandy Mason and other members of the MMRU for constructive critique and technical advice throughout this project. The content hereof is the sole responsibility of the authors and does not necessarily represent the official views of the SAMRC or the funders.

## Additional information

### Funding

| Funder | Grant reference number | Author |
|---|---|---|
| Eunice Kennedy Shriver National Institute of Child Health and Human Development | U01HD085531 | Digby F Warner |
| Norges Forskningsråd | R&D Project 261669 | Digby F Warner |
| South African Medical Research Council | | Timothy J de Wet Valerie Mizrahi Digby F Warner |
| National Research Foundation | | Valerie Mizrahi Digby F Warner |

The funders had no role in study design, data collection and interpretation, or the decision to submit the work for publication.

### Author contributions

Timothy J de Wet, Conceptualization, Resources, Data curation, Software, Formal analysis, Validation, Investigation, Visualization, Methodology, Writing - original draft, Project administration, Writing - review and editing; Kristy R Winkler, Investigation, Methodology, Writing - review and editing; Musa Mhlanga, Conceptualization, Supervision, Visualization; Valerie Mizrahi, Resources, Supervision, Funding acquisition, Writing - review and editing; Digby F Warner, Conceptualization, Resources, Supervision, Funding acquisition, Writing - original draft, Writing - review and editing

### Author ORCIDs

Timothy J de Wet  https://orcid.org/0000-0002-3978-5322
Kristy R Winkler  http://orcid.org/0000-0003-3098-3290
Valerie Mizrahi  http://orcid.org/0000-0003-4824-9115
Digby F Warner  https://orcid.org/0000-0002-4146-0930

### Decision letter and Author response

Decision letter https://doi.org/10.7554/eLife.60083.sa1
Author response https://doi.org/10.7554/eLife.60083.sa2

## Additional files

### Supplementary files

- Supplementary file 1. Key reagents.
- Supplementary file 2. Essentiality calls, functional annotations and identified transcriptional units.
- Supplementary file 3. Literature review of genes targeted in this work.
- Supplementary file 4. Supplementary methods.
- Transparent reporting form

## Data availability

All scripts and underlying data are available via the Open Science Framework at https://osf.io/pdcw2/. All phenotypic data are made available via an interactive database at https://timdewet.shinyapps.io/MorphotypicLanscape/.

The following dataset was generated:

| Author(s) | Year | Dataset title | Dataset URL | Database and Identifier |
|---|---|---|---|---|
| de Wet TJ | 2020 | Arrayed CRISPRi and Quantitative Imaging Describe the Morphotypic Landscape of Essential Mycobacterial Genes | https://osf.io/pdcw2/ | Open Science Framework, pdcw2 |

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
