## [Decision Letter]

**Acceptance summary:**

This manuscript represents important technological and biological advances in our understanding of how different essential pathways in mycobacteria intersect with effects on bacterial morphology, identifying previously unknown functional associations. This study provides a framework for future work that could accelerate our understanding of gene function in mycobacteria.

**Decision letter after peer review:**

Thank you for submitting your article "Arrayed CRISPRi and quantitative imaging describe the morphotypic landscape of essential mycobacterial genes" for consideration by *eLife*. Your article has been reviewed by three peer reviewers, and the evaluation has been overseen by a Reviewing Editor and Gisela Storz as the Senior Editor. The following individual involved in review of your submission has agreed to reveal their identity: E Hesper Rego (Reviewer #3).

The reviewers have discussed the reviews with one another and the Reviewing Editor has drafted this decision to help you prepare a revised submission.

Summary:

This manuscript combines a CRISPRi library in *Mycobacterium smegmatis* with high throughput light microscopy and image analysis to investigate the effects of essential gene knockdown on bacterial morphology. The reviewers all agree that there are many technical advances presented in this paper, the experiments are well executed, and the data and its analysis is significant for the field. However, there are some questions regarding the reproducibility of the data and the utility of these data as a predictive tool. The reviewers believe that these questions should be straightforward to address, as described more below.

1) Questions of reproducibility: The authors state that "Moreover, to verify the reproducibility of the imaging workflow, replicate imaging was performed on separate days for 134 strains."

Does this mean that the authors don't have replicate data for 29 strains? To ensure reproducibility, the authors should perform one or both of the following: 1) Finish collecting the replicate data sets to ensure reproducibility and/or 2) Address reproducibility by comparing the data for the 129 that have been replicated.

The authors should also validate a few genes with a second guide RNA to rule out off-target effects and confirm phenotypes.

2) Questions of utility as a predictive tool:

a) MSMEG_3213 isn't an example of defining the function of an uncharacterized gene instead it simply validates existing database predictions. Further, the data presented here do not demonstrate that MSMEG_3213 is the methylase of an R-M pair. Limitations should be made clear in the discussion of the data.

b) The approach and data falls short of broadly being able to predict the function for any essential gene of interest. The data as presented in Figure 6 do not help this case. While some functionally related genes cluster together, many do not, especially for genes that fall into cluster 2. The disorganization in the UMAP space may stem from the small number of observed phenotypes, whereas published work in other organisms reports much broader ranges of depletion phenotypes. That being said, this isn't a fault of the authors', but it does diminishes their claim to use this method as a predictive tool. The text should be reworded or restructured to clearly represent the utility (or limitations) of these data as a predictive tool.

c) Reshuffling or restructuring some of the sections may help to guide the reader towards understanding the utility of the methods and the data. For example, in addition to describing the methods of their technique, the author's validate or give examples of what their data contain (identification of cryptic putative RM system, histidine auxotroph phenotypes, effects of disrupting mycolic acid biosynthesis). They then discuss the potential to use CRISPRi to confirm compound MOA. This is a lot of information (10 figures with many subpanels), but none of these threads are really taken to completion.

[Editors' note: further revisions were suggested prior to acceptance, as described below.]

Thank you for resubmitting your work entitled "Arrayed CRISPRi and quantitative imaging describe the morphotypic landscape of essential mycobacterial genes" for further consideration by *eLife*. Your revised article has been evaluated by Gisela Storz (Senior Editor) and a Reviewing Editor.

The manuscript has been improved but there are some remaining issues that need to be addressed before acceptance, as outlined below:

1) "An immediate priority is to expand the library to include all 423 predicted essential *M. smegmatis* genes from our pooled screening data (de Wet et al., 2018) and to perform additional replicate imaging for all mutants to enhance resolution and reproducibility."

While this passage might assuage a reviewer or provide the impetus for a sub-Aim of a proposal, it should possibly be omitted because it implies that the missing data are too important to omit and therefore it is premature to publish the current body of work. A stated immediate priority in a published report is acceptable when it describes the next logical step in a progression identified by conclusions of that publication, but it is less acceptable when it describes a gap in the present step. The wording in the passage sounds like the latter.

The subset of genes selected and currently presented is justified on the basis of conservation with *M. tuberculosis*, and the degree of reproducibility demonstrated by an independent replicate of more than half of those suggest high value and reliable data.

2) "Although morphological profiling appears to offer a rapid means of preliminary gene function assignment or compound MOA, this approach cannot claim single gene-level sensitivity. Definitive validation is therefore required – via further biochemical and/or functional analysis – to ratify the functional assignments predicted using this tool."

The phrase "gene-level sensitivity" may be misapplied in this context, or at least, confusing. A reviewer points out that the manuscript demonstrates gene-level sensitivity by reducing expression of a gene and the subsequent combination of morphologic effects (phenoprint) places that gene in the Euclidean neighborhood of other genes. The overall passage is aimed at qualifying a limitation of the screen in that it cannot biochemically discern the basis for its inclusion in the neighborhood. Therefore, a clearer and more concise passage would delete "this approach cannot claim single gene-level sensitivity." The resulting single sentence says that phenoprint data does not assign gene function but gene membership to a process or pathway. This can then guide specific experiments of the functional basis for that membership.

3) "Despite lacking single-gene resolution, morphological profiling has the capacity to identify mutants with unexpected phenotypes, providing a preliminary phenotypic characterization which can guide focused downstream investigations towards assigning gene function."

Same comment about the use of "single-gene". Just starting with "Morphological profiling…" could be a clearer statement.

---

## [Author Response]

Revisions for this paper:1) Questions of reproducibility: The authors state that "Moreover, to verify the reproducibility of the imaging workflow, replicate imaging was performed on separate days for 134 strains."Does this mean that the authors don't have replicate data for 29 strains? To ensure reproducibility, the authors should perform one or both of the following: 1) Finish collecting the replicate data sets to ensure reproducibility and/or 2) Address reproducibility by comparing the data for the 129 that have been replicated.

The reviewers are correct in their interpretation that replicate imaging was performed on 134 (of 263) strains, with an additional 3 added during revision; in addition, 27 replicates were performed of the vector-only control strain. These numbers are clarified in new Figure 1—figure supplement 1. In obtaining replicate imaging data for a subset of strains, we were guided by the approach of Campos et al., 2018, who applied selective or partial replicate imaging to ensure reproducibility in their pioneering study. To avoid confusion, we have clarified this in the manuscript with the following statements:

“To confirm the reproducibility of data obtained from the single-timepoint, single-replicate imaging workflow, we selected 137 strains for re-imaging on separate days (Figure 1—figure supplement 1).”

The strong correlation we observed between replicates instilled confidence in the single-replicate imaging strategy, allowing us to adopt this approach for the remaining strains, all of which represent data obtained from a median of 400 single cells. We nevertheless acknowledge the desirability of including additional replicate imaging sets for the remaining strains and, when local conditions allow for a return to work post COVID-19 lockdown, we will perform the additional replicates and update our online data repository accordingly. We have added the following line in the Discussion to confirm this intention:

“An immediate priority is to expand the library to include all 423 predicted essential *M. smegmatis* genes from our pooled screening data (de Wet et al., 2018) and to perform additional replicate imaging for all mutants to enhance resolution and reproducibility.”

The authors should also validate a few genes with a second guide RNA to rule out off-target effects and confirm phenotypes.

The reviewers are correct in identifying off-target effects as a pervasive concern with CRISPRi-based approaches (Cui, Vigouroux, Rousset, et al., 2018 Nat Commun **9,** 1912 doi: 10.1038/s41467-018-04209-5). To mitigate the impact of off-target effects, we utilized two approaches previously in constructing our genome-scale library (de Wet et al., 2018) and, therefore, in developing the ranking of guide efficacies which informed guide selection here: first, we applied an *in silico* approach for guide selection that was optimized to avoid close homology; and, secondly, empirical data on individual sgRNA efficacies – inferred from quantitative analyses of mutant abundances – were utilized to inform guide choice in the current study.

We also agree that, ideally, each phenotype should be validated with additional guide RNAs. However, this becomes impractical at scale and, in evaluating our experimental approach, we were reassured by:

i) literature precedent (*e.g*., Liu et al., 2017) supporting the use of single mutant phenotypes;

ii) the observation that, through a comprehensive literature review, we were able to align most reported phenotypes (where available) with those described herein; and

iii) the consistency of phenotypes observed for genes contained within operons, where polar effects are most predicted (Figure 5—figure supplement 1) – in essence, different guides targeting the same transcriptional unit produce the same phenotype.

Notwithstanding the above, we had data available from previous validations of a small panel of genes; these results are now included in new Figure 2—figure supplement 2, which confirms the consistency of the reported phenotypes with additional sgRNAs.

2) Questions of utility as a predictive tool:a) MSMEG_3213 isn't an example of defining the function of an uncharacterized gene instead it simply validates existing database predictions. Further, the data presented here do not demonstrate that MSMEG_3213 is the methylase of an R-M pair. Limitations should be made clear in the discussion of the data.

We thank the reviewers for this comment, which rightly cautions against overinterpreting the utility of this approach. It is correct that the functionality of MSMEG_3213 was predicted in the REBASE database, although we submit that the existing annotation was without functional validation and, moreover, was not contained in databases commonly accessed by mycobacteriologists; for example, Mycobrowser (Kapopoulou et al., 2011; Tuberculosis (Edinb). 91, 8-13).

We also agree that while our data are strongly suggestive of methylase function, they are not definitive. Additional in vitro biochemical data validating protein function, or in vivo data confirming a decrease in DNA methylation consequent on knockdown, are required to establish unequivocally that MSMEG_3213 is a methylase. Our results provide the impetus for the proposed follow-up studies.

In acknowledging the caveats outline above, we have inserted the following statement in the revised manuscript:

“Further biochemical and/or functional characterization is required before MSMEG_3213 can be definitively assigned as methylase; however, the evidence derived here from morphological profiling, transcriptomic profiling and combinatorial CRISPRi strongly support the identification of a predicted Type II R-M system in *M. smegmatis*.”

In addition, we have modified the Abstract as follows:

“Leveraging statistical-learning, we demonstrate that functionally related genes cluster by morphotypic similarity and that this information can be used to inform investigations of gene function. Exploiting this observation, we infer the existence of a mycobacterial restriction-modification system, and identify filamentation as a defining mycobacterial response to histidine starvation.”

b) The approach and data falls short of broadly being able to predict the function for any essential gene of interest. The data as presented in Figure 6 do not help this case. While some functionally related genes cluster together, many do not, especially for genes that fall into cluster 2. The disorganization in the UMAP space may stem from the small number of observed phenotypes, whereas published work in other organisms reports much broader ranges of depletion phenotypes. That being said, this isn't a fault of the authors', but it does diminishes their claim to use this method as a predictive tool. The text should be reworded or restructured to clearly represent the utility (or limitations) of these data as a predictive tool.

We thank the reviewers for this comment which again stresses the need for prudence in interpreting the utility of this approach. In acknowledging this caution, we have updated the Discussion with the following statement:

“Although morphological profiling appears to offer a rapid means of preliminary gene function assignment or compound MOA, this approach cannot claim single gene-level sensitivity. Definitive validation is therefore required – via further biochemical and/or functional analysis – to ratify the functional assignments predicted using this tool.”

In addition, we have included additional text noting these limitations within the Results section of the paper, adding:

“Despite lacking single-gene resolution, morphological profiling has the capacity to identify mutants with unexpected phenotypes, providing a preliminary phenotypic characterization which can guide focused downstream investigations towards assigning gene function.”

c) Reshuffling or restructuring some of the sections may help to guide the reader towards understanding the utility of the methods and the data. For example, in addition to describing the methods of their technique, the author's validate or give examples of what their data contain (identification of cryptic putative RM system, histidine auxotroph phenotypes, effects of disrupting mycolic acid biosynthesis). They then discuss the potential to use CRISPRi to confirm compound MOA. This is a lot of information (10 figures with many subpanels), but none of these threads are really taken to completion.

We thank the reviewers for this comment which echoes our own deliberations in how best to present both the methodology and its potential uses. In line with the reviewers’ suggestion, we have extended and adapted the scope of the section on the putative RM system to include additional examples of utility, specifically:

“Other examples supporting the utility of morphological profiling to inform single-gene functional analyses arose during the course of this work (Figure 7—figure supplement 1). For example, the *C. glutamicum* homolog of *MSMEG_0317* is required for lipomannan maturation and lipoarabinomannan (LM/LAM) synthesis (Cashmore et al., 2017). We observed that *MSMEG_0317* clustered closely with genes involved in arabinogalactan synthesis and localized to the cell wall (Figure 7—figure supplement 1). It was pleasing, therefore, when a separate study emerged suggesting that *MSMEG_0317* was involved in the transport of LM/LAM (Gupta et al., 2019). In another example, the transcriptional regulator, *whiA*, which is involved in sporulation in *Streptomyces* (Bush et al., 2013), appears to play a key role in the mycobacterial cell cycle, clustering with other components of cell division (Figure 7—figure supplement 1). Furthermore, *MSMEG_6276* – a putative mur ligase – clusters closely with the peptidoglycan synthesis protein, *mviN* (Gee et al., 2012), and exhibits strong homology to *murT/gatD* from *S. pneumoniae* (Morlot et al., 2018) (Figure 7—figure supplement 1). In combination, these additional examples support the utility of image-based profiling for informing or validating hypothetical or predicted gene function – particularly those involved in cell wall metabolism or DNA metabolism and cell cycle regulation.”

While this does not take any thread “to completion”, we have attempted to emphasize each of the biological examples (identification of cryptic putative RM system, histidine auxotroph phenotypes, effects of disrupting mycolic acid biosynthesis, and MOA determination) as illustrative of utility, rather than composite studies in-and-of themselves. Again, our intention here is to demonstrate the capacity for morphotypic phenotypes – especially where compared against a database of other gene-specific morphotypes – to provide the impetus for downstream analyses which might yield functional information. That is, as a key tool in functional genomics.

[Editors' note: further revisions were suggested prior to acceptance, as described below.]

The manuscript has been improved but there are some remaining issues that need to be addressed before acceptance, as outlined below:1) "An immediate priority is to expand the library to include all 423 predicted essential M. smegmatis genes from our pooled screening data (de Wet et al., 2018) and to perform additional replicate imaging for all mutants to enhance resolution and reproducibility."While this passage might assuage a reviewer or provide the impetus for a sub-Aim of a proposal, it should possibly be omitted because it implies that the missing data are too important to omit and therefore it is premature to publish the current body of work. A stated immediate priority in a published report is acceptable when it describes the next logical step in a progression identified by conclusions of that publication, but it is less acceptable when it describes a gap in the present step. The wording in the passage sounds like the latter.The subset of genes selected and currently presented is justified on the basis of conservation with M. tuberculosis, and the degree of reproducibility demonstrated by an independent replicate of more than half of those suggest high value and reliable data.

This point is valid and well made. As recommended in the editorial comment, the text has been edited as follows in the revised version:

“To ensure completeness in our online “Morphotypic Landscape” database, we plan in future to expand the library to include all 423 predicted essential *M. smegmatis* genes from our pooled screening data (de Wet et al., 2018).”

2) "Although morphological profiling appears to offer a rapid means of preliminary gene function assignment or compound MOA, this approach cannot claim single gene-level sensitivity. Definitive validation is therefore required – via further biochemical and/or functional analysis – to ratify the functional assignments predicted using this tool."The phrase "gene-level sensitivity" may be misapplied in this context, or at least, confusing. A reviewer points out that the manuscript demonstrates gene-level sensitivity by reducing expression of a gene and the subsequent combination of morphologic effects (phenoprint) places that gene in the Euclidean neighborhood of other genes. The overall passage is aimed at qualifying a limitation of the screen in that it cannot biochemically discern the basis for its inclusion in the neighborhood. Therefore, a clearer and more concise passage would delete "this approach cannot claim single gene-level sensitivity." The resulting single sentence says that phenoprint data does not assign gene function but gene membership to a process or pathway. This can then guide specific experiments of the functional basis for that membership.

Again, this point is fair and well made. Moreover, the edit suggested following editorial review nicely accomplishes that which our own previous version failed to do: that is, convey the fact that the limitation of the morphological profiling approach inheres in the fact that it cannot biochemically discern the basis for the inclusion of a specific gene in a given Euclidean neighborhood. As recommended, the text has been edited as follows in the revised version:

“Morphological profiling appears to offer a rapid means of preliminary gene function assignment or compound MOA; however, definitive validation is required – via further biochemical and/or functional analysis – to ratify the functional assignments predicted using this tool.”

3) "Despite lacking single-gene resolution, morphological profiling has the capacity to identify mutants with unexpected phenotypes, providing a preliminary phenotypic characterization which can guide focused downstream investigations towards assigning gene function."Same comment about the use of "single-gene". Just starting with "Morphological profiling…" could be a clearer statement.

As acknowledged above, this is a valid point and the suggested edit – removing the phrase “Despite lacking single-gene resolution” – resolves this problem. The revised version reflects this change:

“Morphological profiling has the capacity to identify mutants with unexpected phenotypes, providing a preliminary phenotypic characterization which can guide focused downstream investigations towards assigning gene function.”